# NeuroGauss4D-PCI: 4D Neural Fields and Gaussian Deformation Fields for Point Cloud Interpolation

**Chaokang Jiang**[1*], **Dalong Du**[1*], **Jiuming Liu**[2], **Siting Zhu**[2],
**Zhenqiang Liu**[1], **Zhuang Ma**[1], **Zhujin Liang**[1], **Jie Zhou**[3†]
[1]PhiGent Robotics, [2]Shanghai Jiaotong University, [3]Tsinghua University [*]

## Abstract

Point Cloud Interpolation confronts challenges from point sparsity, complex spatiotemporal dynamics, and the difficulty of deriving complete 3D point clouds from sparse temporal information. This paper presents NeuroGauss4D-PCI, which excels at modeling complex non-rigid deformations across varied dynamic scenes. The method begins with an iterative Gaussian cloud soft clustering module, offering structured temporal point cloud representations. The proposed temporal radial basis function Gaussian residual utilizes Gaussian parameter interpolation over time, enabling smooth parameter transitions and capturing temporal residuals of Gaussian distributions. Additionally, a 4D Gaussian deformation field tracks the evolution of these parameters, creating continuous spatiotemporal deformation fields. A 4D neural field transforms low-dimensional spatiotemporal coordinates $(x, y, z, t)$ into a high-dimensional latent space. Finally, we adaptively and efficiently fuse the latent features from neural fields and the geometric features from Gaussian deformation fields. NeuroGauss4D-PCI outperforms existing methods in point cloud frame interpolation, delivering leading performance on both object-level (DHB) and large-scale autonomous driving datasets (NL-Drive), with scalability to auto-labeling and point cloud densification tasks. The source code is released at github.com/jiangchaokang/NeuroGauss4D-PCI.

## 1 Introduction

Point cloud frame interpolation (PCI) [1; 2] aims to estimate intermediate frames given two or more point cloud frames. This task enables the generation of temporally smooth and continuous point cloud sequences at arbitrary timestamps, which is crucial for applications such as autonomous driving [3; 4] and virtual reality [5; 6; 7]. PCI can be expressed with the following formula:

$$\mathcal{F}_\Theta(\{\underbrace{\mathbf{P} \in \mathbb{R}^{N_i \times 3}\}_{t=0,4,8\cdots}, T_{t=0,4,8\cdots}}_{\text{Training Data}}), \qquad \underbrace{\mathcal{F}_\Theta(\mathbf{P}_{t=i}, T_{t=i}, T_{t=j})}_{\text{Inference Input}} \rightarrow \{\mathbf{P}_{t=j}^{Pred} \in \mathbb{R}^{N \times 3}\}. \quad (1)$$

PCI faces several challenges due to the unique characteristics of point cloud data and the complexity of modeling spatiotemporal dynamics: **1)** Point clouds are inherently sparse and unordered, lacking the regular structure of images. For instance, NeuralPCI [3] simply concatenates spatial and temporal coordinates as inputs to an MLP, struggling to adequately represent the motion and correlation of multiple unordered point clouds over time. **2)** PCI involves modeling the spatiotemporal dynamics of point clouds, requiring the interpolation model $\mathcal{F}_\Theta$ to capture the spatial structure and temporal evolution of the scene from 4D training data ($\{\mathbf{P} \in \mathbb{R}^{N_i \times 3}\}_{t=0,4,8\cdots}, T_{t=0,4,8\cdots}$), and model the non-rigid deformations and non-linear trajectories of discrete 3D points. This leads to the linear motion assumption of PointINet [8], which uses bidirectional scene flow to warp input frames for

---

[*]Equal contribution. † Corresponding author.

38th Conference on Neural Information Processing Systems (NeurIPS 2024).

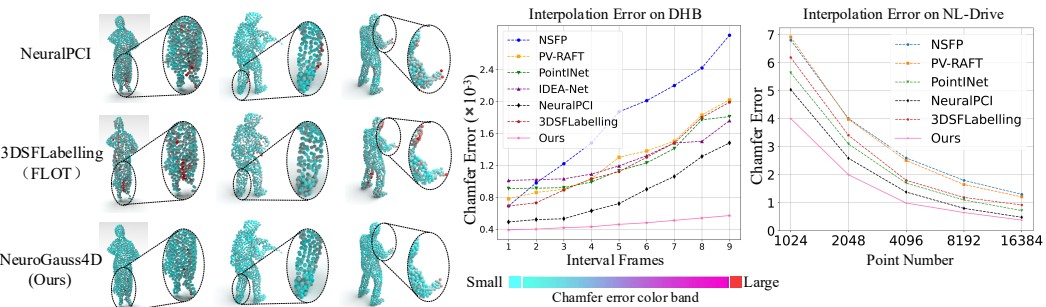

Figure 1: NeuroGauss4D-PCI robustly outperforms existing methods [3; 9] across multiple datasets, frame intervals, and point cloud densities, consistently achieving lower interpolation errors. NeuroGauss4D-PCI robustly handles minute non-rigid deformations, large-scale unstructured scenes, dynamic environments with non-uniform data, and extensive motions. The proposed method consistently achieves precise local and global point cloud predictions.

estimating intermediate frames, failing to capture complex non-linear motion. Moreover, 3D scene flow estimation methods, such as 3DSFLabelling [9], PV-RAFT [10], and NSFP [11], can only express the motion field between two frames but cannot robustly represent higher-order dynamic scenes over longer time spans. **3)** The inference process in PCI faces the challenge of generalizing from sparse temporal samples. The model $\mathcal{F}_\Theta$ generates an accurate point cloud $\mathbf{P}_{t=j}^{Pred} \in \mathbb{R}^{N \times 3}$ for time $t = j$ based on $P_{t=i}$ and time frame at $T_{t=i}$ and the time frame at $T_{t=j}$. This demands strong interpretability and 4D modeling capabilities from the model to accurately predict the interpolated point cloud from minimal information. IDEA-Net [2] addresses the correlation between two input frames by learning a one-to-one alignment matrix and refining the linear interpolation results using a trajectory compensation module. However, for large-scale, occluded autonomous driving point cloud scenes, IDEA-Net's one-to-one correspondence assumption struggles to accurately predict the point cloud at interpolated timestamps from sparse temporal samples.

For the challenges of point cloud frame interpolation, we propose a novel 4D spatio-temporal modeling method based on Gaussian representations, NeuroGauss4D-PCI. It first captures the geometric structure of point clouds through iterative Gaussian soft clustering, providing structured representations for unordered data. Then, the temporal radial basis function Gaussian residual module interpolates over discrete time steps, learning the continuous dynamics of Gaussian parameters to capture the spatio-temporal evolution of point clouds. The 4D Gaussian Deformation Field further models the spatio-temporal variation trends of Gaussian parameters, effectively overcoming the limitations of linear interpolation. Finally, the attention fusion module adaptive fusion of latent neural representations and explicit geometric representations, enhancing the modeling capability of spatio-temporal correlations. As shown in Fig. 1, this compact and efficient method demonstrates superior point cloud sequence generation performance on multiple datasets.

**Contributions:** • We propose NeuroGauss4D-PCI, a novel 4D spatio-temporal modeling method for point cloud frame interpolation. It captures geometric structures through iterative Gaussian soft clustering, providing structured representations for unordered point clouds, while adaptively fusing the latent neural field features. • A Temporal Radial Basis Function Gaussian Residual module is designed to interpolate over discrete time steps, learning continuous Gaussian parameter dynamics to effectively model the spatio-temporal evolution of point clouds. • An innovative 4D Gaussian deformation field is introduced to model the spatio-temporal variations of Gaussian parameters, achieving smooth and realistic point cloud deformations through 4D Gaussian graph convolutions and Gaussian representation pooling. • Our method outperforms others on multiple datasets, generating accurate point cloud sequences with minimal errors. It can be easily extended to tasks like lidar-camera temporal synchronization and point cloud densification.

## 2   Related Work

**Neural Field**   In computational geometry, latent neural fields [12; 11; 13] encode complex 3D scenes with the principle $f_\Theta(\phi(\chi)) = \mathbf{o}$. $\chi$ denotes the input parameters, such as coordinates in a 3D space. The function $\phi$ is a feature transformation applied to $\chi$, which enhances the neural network's ability to model complex patterns by providing a high-dimensional representation of the input space. The neural network $f_\Theta$, parameterized by $\Theta$, then maps these transformed features to the output $\mathbf{o}$.

This output can represent various aspects of the 3D scene, such as color [14], density [15], or surface normals [16], depending on the task. GPNF [17] presents a method using neural fields for efficient and topology-flexible geometry processing, outperforming traditional mesh methods in shape editing tasks. NSFP [11] introduces a novel implicit regularizer based on neural scene flow priors, utilizing coordinate-based MLPs for robust scene flow optimization. Implicit neural representations [18] ensure differentiable interpolation for 3D point clouds, facilitating end-to-end learning [19], yet their performance is limited by a dependency on high-quality training data [9] and computational demands [20], with inherent trade-offs in interpretability.

**3D Gaussian Splatting**   Recent studies have demonstrated that 3D Gaussian Splatting (3D GS) [21; 22; 23] leverages an explicit radiance field for efficient:

$$I(p) = \sum_{i=1}^{N} w_i \cdot G(p; \mu_i, \Sigma_i) = \sum_{i=1}^{N} w_i \cdot \exp((-\frac{1}{2}(p - \mu_i)^T \Sigma_i^{-1}(p - \mu_i)), \quad (2)$$

where $I(p)$ represents the rendered intensity or color at pixel location $p$, $N$ is the total number of 3D Gaussian models in the scene, $w_i$ is the weight of the $i$-th Gaussian model affecting its contribution to the final image, $G(p; \mu_i, \Sigma_i)$ is the 3D Gaussian function defined as $\exp\left(-\frac{1}{2}(p - \mu_i)^T \Sigma_i^{-1}(p - \mu_i)\right)$, $\mu_i$ is the center of the Gaussian representing the position in 3D space, and $\Sigma_i$ is the covariance matrix that determines the shape and scale of the Gaussian distribution. 3DGS [21] innovatively employs 3D Gaussians to represent scenes, optimizing volumetric radiance fields with minimal computational waste, enabling high-quality real-time novel-view synthesis. Recent works [24; 25; 26] focused on creating 4D scene representations using Gaussian splatting from video to model geometry and appearance changes across frames. Few explored modeling temporal 3D motion changes from raw, sparse 3D point clouds, requiring tracking spatial coordinates over time and dynamically adjusting Gaussian parameters to accurately reflect temporal deformations.

**Point Cloud Interpolation (PCI)**   Existing PCI models are categorized into three types: based on 3D scene flow [10; 20; 27; 8; 28], neural fields[11; 3], and feature alignment [29] or trajectory regression [2]. 3D scene flow methods estimate correspondences between frames and use linear interpolation for intermediate frames. PointINet [8] calculates bidirectional 3D scene flow to warp point clouds and insert new frames. IDEA-Net [2] applies aligned feature embeddings to manage trajectory nonlinearities, enhancing interpolation. NeuralPCI [3] implicitly learns the nonlinear temporal motion characteristics of multi-frame point clouds through a neural field. However, these methods struggle with modeling 4D geometric deformations in point cloud sequences, impacting robustness. Our method combines 4D neural fields and Gaussian deformation fields to effectively model and predict dynamic point cloud changes, capturing nonlinear motion and deformation accurately.

## 3   The Algorithm

### 3.1   Overview of 4D Neural Field And Gaussian Deformation Field

As depicted in Figure 2, temporal point cloud coordinates are encoded using Fourier basis functions to encapsulate periodic information as follows:

$$Posenc(\mathbf{P}|(\mathbf{x}, \mathbf{y}, \mathbf{z}, \mathbf{t})) = [\Psi(\mathbf{x}), \Psi(\mathbf{y}), \Psi(\mathbf{z}), \Psi(\mathbf{t})]; \Psi(\mathbf{x}) = [\mathbf{x}, \sin(\mathbf{x}), \cos(\mathbf{x})]. \quad (3)$$

To enhance the modeling capability for periodic features, the original coordinate features are augmented by incorporating their sine and cosine values. Features from $Posenc(\mathbf{P}|(\mathbf{x}, \mathbf{y}, \mathbf{z}, \mathbf{t}))$ are further encoded using a 4D neural field. The iterative Gaussian cloud soft clustering module employs Gaussian statistics to extract characteristics such as mean, covariance and geometric features from the point cloud, offering a statistical description of internal dynamic changes. Furthermore, the proposed temporal radial basis function Gaussian residual (RBF-GR) module employs radial basis functions to interpolate these Gaussian parameters across different time steps. The designed 4D Gaussian deformation field module learns the temporal evolution of point cloud Gaussian parameters, generating a continuous spatiotemporal deformation field. Finally, an fast latent-geometric fusion module adaptively fusion features, enabling the generation of point clouds through a prediction head.

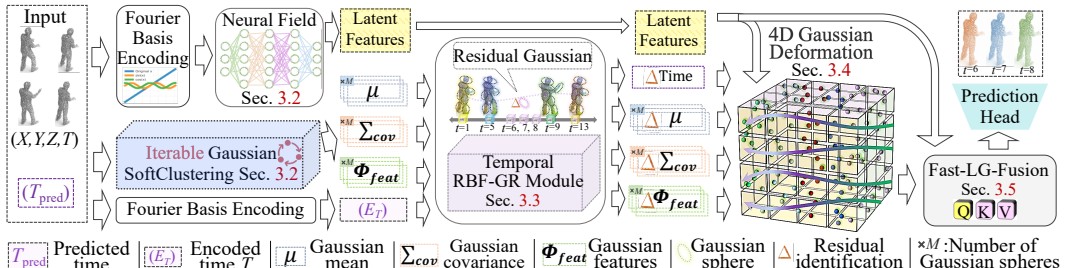

Figure 2: Three key steps of NeuroGauss4D-PCI: **1)** Latent feature learning and point cloud Gaussian representation: Fourier feature mappings and 4D neural fields map low-dimensional temporal coordinates $(x, y, z, t)$ to high-dimensional latent features, while representing the original temporal point clouds as robust multi-Gaussian ellipsoids $(\mu, \Sigma, \Phi)$. **2)** The temporal radial basis function Gaussian residual (RBF-GR) module captures residuals among temporal Gaussian distributions, fusing smooth temporal Gaussian distributions with latent features to construct a 4D Gaussian deformation field that learns and smoothens point cloud deformations. **3)** An efficient transformer architecture aggregates features from the 4D deformation field and latent features, enabling point cloud interpolations at any given timestamp through a point cloud prediction head.

## 3.2 Iterative Gaussian Soft Clustering and 4D Neural Field

**Algorithm 1** Iterable Gaussian Soft Clustering

---

**Require:** $\mathbf{P} \in \mathbb{R}^{N \times 3}$: point coordinates, $M$: number of Gaussians, $\kappa$: number of iterations
**Ensure:** $\boldsymbol{\mu} \in \mathbb{R}^{M \times 3}$: Gaussian Means, $\boldsymbol{\Sigma} \in \mathbb{R}^{M \times 3 \times 3}$: Gaussian Covariances Matrix, $\Phi_{feat}^G$: Gaussian features, $\mathbf{A} \in \mathbb{R}^N$: Point-to-Gaussian Index Matrix
1: Initial centers matrix $\mathbf{C} \in \mathbb{R}^{M \times 3}$ by randomly selecting $M$ points from $\mathbf{P}$
2: **for** $\tau = 1$ to $\kappa$ **do**
3: $\quad \mathbf{D}_{ij} = \|\mathbf{P}_i - \mathbf{C}_j\|_2, \mathbf{D} \in \mathbb{R}^{N \times M}$
4: $\quad \mathbf{S}_{ij} = \frac{\exp(-\mathbf{D}_{ij}^2)}{\sum_{m=1}^M \exp(-\mathbf{D}_{im}^2)}, \mathbf{S} \in \mathbb{R}^{N \times M}$
5: $\quad$ Update $\mathbf{C} = (\mathbf{S}^\top \mathbf{P}) \oslash (\mathbf{S}^\top \mathbf{1} + \epsilon)$
6: **end for**
7: $\boldsymbol{\mu} = \mathbf{C}$
8: $\mathbf{A}_i = \arg\max_j \mathbf{S}_{ij}, \forall_i \in \{1, \ldots, N\}$
9: **for** $m = 1$ to $M$ **do**
10: $\quad \mathbf{P}_m = \mathbf{P}[\mathbf{A} == m] - \boldsymbol{\mu}_m$
11: $\quad \boldsymbol{\Sigma}_m = \frac{1}{|\mathbf{P}_m|} \sum_{\mathbf{p} \in \mathbf{P}_m} \mathbf{p}\mathbf{p}^\top + \epsilon \mathbf{I}_3$
12: **end for**
13: **for** $i = 1$ to $M$ **do**
14: $\quad \mathbf{F}^G = Dgcnn(\mathbf{P}[\mathbf{A} == i])$ $\qquad \triangleright$ Eq. 4.
15: $\quad \mathbf{F}_{att}^G = SelfAttention(\mathbf{F}^G)$ $\qquad \triangleright$ Eq. 5.
16: $\quad \Phi_{feat}^G$.append$(\mathbf{F}_{att}^G)$
17: **end for**

---

The iterative Gaussian cloud soft-clustering representation, given input point coordinates $\mathbf{P}$, number of Gaussians $M$, consists of three steps: **1)** Soft-clustering of point clouds to Gaussian distributions is achieved by introducing soft assignment weights $\mathbf{S}_{ij}$, where $\mathbf{S}_{ij}$ represents the probability of the $i$-th point belonging to the $j$-th Gaussian, calculated based on the Euclidean distances $\mathbf{D}_{ij}$ between points and Gaussian centers $\mathbf{C}$, overcoming the limitations of traditional hard-clustering methods. **2)** The Gaussian means $\boldsymbol{\mu}$ and covariance matrices $\boldsymbol{\Sigma}$ are iteratively optimized over $\kappa$ iterations, with $\boldsymbol{\mu}$ updated by minimizing the distances $\mathbf{D}_{ij}$ and $\boldsymbol{\Sigma}$ calculated based on the distribution of points $\mathbf{P}_m$ relative to the Gaussian means, adaptively fitting the spatial distribution of point clouds. **3)** Multi-scale Gaussian features $\Phi_{feat}^G$ are extracted by applying DGCNN [30] and self-attention mechanisms to points assigned to each Gaussian, learning rich local and global features. Finally, the point cloud is assigned to the corresponding Gaussians through the index matrix $\mathbf{A}$.

The raw point cloud coordinates lack topological structure and contain noise and redundancy. We propose an iterative Gaussian soft-clustering point cloud representation module (Algorithm 1) that converts point clouds into structured Gaussian representations. The module introduces DGCNN [30] (Eq. 4) to extract local geometric features, where $\mathbf{x}_i$ is the $i$-th point, $\mathbf{x}_j$ is its k-nearest neighbor, $\Gamma_{graph}$ is the graph convolution operation. $\theta g$ is the convolution parameter. $\uplus$ represents tensor concatenation; Self-Attention (Eq. 5) to capture global contextual information, where $\mathbf{W}_Q$, $\mathbf{W}_K$, $\mathbf{W}_V$ are the weight parameters of the query, key, and value matrices, respectively, and $d_k$ is the dimension of the key matrix.

$$\mathbf{F}^G = \left\{ \mathbf{x}_i' = \max_{j \in \mathbf{KNN}(i,k)} \sigma_{Relu} \left( \Gamma_{graph} \left( ((\mathbf{x}_i - \mathbf{x}_j) \uplus \mathbf{x}_i); \theta_g \right) \right) \ \forall_i \in \mathbf{P} \right\}, \tag{4}$$

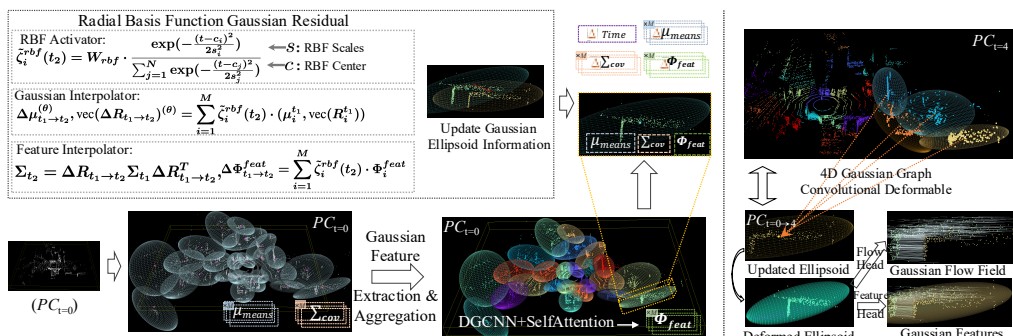

Figure 3: Temporal radial basis function Gaussian residual (RBF-GR) and 4D Gaussian Deformation Field. The normalized RBF weights $\tilde{\zeta}_i^{rbf}(t_2)$ are used to compute the residuals of Gaussian means $\Delta\mu_{t_1\to t_2}^{(\theta)}$, rotations $\Delta R_{t_1\to t_2}^{(\theta)}$, and features $\Delta\Phi_{t_1\to t_2}^{feat}$ between time $t_1$ and $t_2$. The covariance matrix $\Sigma_{t_2}$ is then updated using the learned rotation residual. $W_{rbf} = \sigma_{softmax}(MLP(\Phi_{t1}^{feat}))$, where $W_{rbf}$ denotes the attention weights for RBF activations, adaptively adjusted based on Gaussian features. The Temporal-RBF-GR employs radial basis functions (RBFs) with learnable centers $c_i$ and scales $s_i$ to capture the temporal evolution of Gaussian parameters.

$$\mathbf{F}_{\text{att}}^{G} = \left(\mathbf{F}^G \mathbf{W}_V\right) \sigma_{Softmax}\left(\frac{(\mathbf{F}^G\mathbf{W}_Q)(\mathbf{F}^G\mathbf{W}_K)^\top}{\sqrt{d_k}}\right)^\top. \tag{5}$$

Except for Gaussian cloud representation, we introduce 4D neural fields, parameterizing a continuous, latent spatio-temporal field function using multi-layer perceptron (MLP) networks, as shown in Fig. 2. This maps low-dimensional spatio-temporal coordinates $(x, y, z, t)$ to a high-dimensional latent feature space, representing the motion and changes of point clouds in the spatio-temporal domain. Compared to Gaussian point cloud representations, deep networks can fit more complex non-linear spatio-temporal correlations.

Point cloud data from LiDAR includes noise, which, while not affecting subsequent perception tasks, impedes the learning of temporal consistency. We apply statistical outlier removal [31] during preprocessing to eliminate noise by calculating each point's average distance to its neighbors and identifying outliers using global mean and standard deviation. Experiments show (Table 2) this method markedly decreases interference during network training.

### 3.3 Temporal Radial Basis Function Gaussian Residual (RBF-GR) Module

This module accomplishes the interpolation and updating of Gaussian distribution parameters in the continuous time domain, which can be primarily divided into three parts (Fig. 3): **RBF Activator:** The RBF Activator converts discrete time steps into continuous time representation by mapping scalar time $t$ to radial basis function activation values $\tilde{\zeta}_i^{rbf}(t)$. RBF networks encode time locally, with each kernel function centered at a specific time $c_i$ and exponentially decaying activations. The RBF activation values $\tilde{\zeta}_i^{rbf}(t)$ reflect the similarity between time $t$ and center times $c_i$, enabling smooth interpolation of Gaussian parameters. **Gaussian Interpolator:** The Gaussian Interpolator interpolates the mean $\mu_t$ and rotation matrix $R_t$ of the Gaussian distribution using RBF activation values. Gaussian parameters estimated at discrete time steps are interpolated to obtain parameter residuals $\Delta\mu_{t_1\to t_2}^{(\theta)}$ and $\Delta R_{t_1\to t_2}^{(\theta)}$ from time $t_1$ to $t_2$. The interpolation process exploits the linearity and closure property of Gaussian distributions. **Feature Interpolator:** The Feature Interpolator interpolates Gaussian features using RBF activation values to obtain a continuous feature representation in time. The module enables smooth updating of Gaussian distribution parameters in continuous time, allowing effective fitting and generation of time-varying 3D point cloud sequences.

### 3.4 4D Gaussian Deformation Field

The temporal Gaussian graph convolutional (TG-GCN) deformation field plays a key role in capturing the spatiotemporal features from Gaussian point cloud representations (Fig. 4). It takes as input

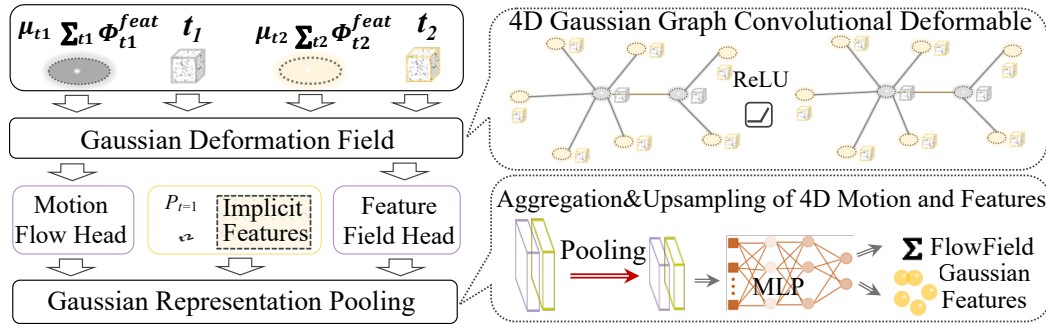

Figure 4: 4D Gaussian Deformation Field. Utilizing Gaussian means $\mu$, covariances $\Sigma$, and features $\Phi^{feat}$, and time $(t_1, t_2, \cdots)$ as inputs, the spatio-temporal graph convolutional network captures spatio-temporal patterns to learn continuous 4D deformation fields. The Gaussian Representation pooling module projects the point cloud onto the Gaussian ellipsoids and upsamples, using max-pooling to extract salient features while capturing complex point dynamics and temporal evolution.

the Gaussian means $\mu$, covariances $\Sigma$, and features $\Phi^{feat}$ extracted from the time-series radial basis function Gaussian residual module, extracts the geometric topological features of local and global Gaussian clouds through the TG-GCN structure, effectively capturing geometric and temporal patterns. The information on **Gaussian means** $\mu$ aids in encoding the average positions $(x, y, z)$ of points at different time steps, modeling overall motion and deformation trends; The information on **Gaussian covariances** $\Sigma$ describe the deformation degree and direction of each point over time, facilitating the modeling of complex dynamics; The **Gaussian features** $\Phi^{feat}$ extract the geometric and temporal patterns in the point cloud, enhancing the model's expressive power and providing geometric features. Furthermore, the incorporation of temporal information aids in establishing the temporal consistency relationships within the 4D spatiotemporal point cloud.

Based on the features extracted from the deformation field, the **motion field head** and **deformation field feature head** respectively predict the motion flow and feature field (Fig. 4), with the core principle being the learning of a continuous deformation field, utilizing the time information in the Gaussian point cloud representation and the spatial dependencies captured by graph convolution, to achieve smooth and realistic deformation of the point cloud, enabling accurate modeling of 4D point cloud sequences and interpolation between time steps.

The Gaussian representation pooling module first projects the original point cloud coordinates and latent features onto Gaussian ellipsoids centered at each point, leveraging the previously established index mapping between the points and the Gaussian ellipsoids (Sec. 3.2). It then applies max-pooling to extract the most prominent features within each Gaussian ellipsoid, enhancing the model's perception of deformation flow and features. However, this Gaussian ellipsoid projection and max-pooling operation inevitably leads to a certain degree of information loss. To compensate for the potential information loss in the feature extraction process of the Gaussian representation pooling module, we further introduce a neural field module to learn and refine the point cloud deformation flow and feature expressions.

### 3.5 Fast Latent-Geometric Feature Fusion (Fast-LG-Fusion) Module

The proposed module fuses the complementary advantages of the latent features $\mathcal{F}_{\mathcal{L}}$ from point cloud latent representations and the geometric features $\mathcal{F}_{\mathcal{G}}$ from 4D Gaussian deformation fields via an attention-based mechanism. $\mathcal{F}_{\mathcal{L}}$ serves as the query, while $\mathcal{F}_{\mathcal{G}}$ serves as the key and value. Through linear projections $(\mathbf{W}_q, \mathbf{W}_k, \mathbf{W}_v)$, the features are mapped into a shared latent space. An attention score is computed between the projected query and key, which is then normalized via softmax to obtain attention weights. These weights adaptively assign importance to different feature components, enabling effective fusion. The weighted sum of the projected value features is then computed, integrating multi-modal information. Finally, a residual connection with the original query

$\mathcal{F}_\mathcal{L}$ yields the fused representation $\mathcal{F}_{\widetilde{\mathcal{L}\&\mathcal{G}}}$, preserving temporal information:

$$\mathcal{F}_{\widetilde{\mathcal{L}\&\mathcal{G}}} = \mathcal{F}_\mathcal{L} + \sigma_{Softmax}\left(\frac{\mathbf{W}_q\mathcal{F}_\mathcal{L} \cdot (\mathbf{W}_k\mathcal{F}_\mathcal{G})^\top}{\sqrt{d_h}}\right)(\mathbf{W}_v\mathcal{F}_\mathcal{G}). \tag{6}$$

The module's efficiency stems from its simple linear operations and avoidance of iterative or recurrent mechanisms, enabling fast computation. Furthermore, the global attention mechanism provides a comprehensive receptive field, enhancing representational capacity compared to naive concatenation. By adaptively fusing geometric and temporal features, the module improves the model's expressiveness for capturing complex spatio-temporal patterns in large-scale dynamic LiDAR scenes.

### 3.6 3D Point Cloud Prediction Head

The 3D Point Cloud Prediction head accurately predicts the point cloud at the target time step by fusing multi-source spatio-temporal information, including features $\mathcal{F}_{\widetilde{\mathcal{L}\&\mathcal{G}}}$ from the 4D neural field and Gaussian deformation field, time encoding $Posenc(T)$ representing the target time step, and the initial prediction $flow + \mathbf{P}$. This fusion provides rich spatio-temporal priors, enabling the model to reason about the target point cloud's structure and motion patterns. Through a lightweight MLP, the fused features are directly mapped to the residual flow $\Delta f^*$, avoiding prediction from scratch and leveraging the initial interpolated flow field based on the Gaussian deformation field:

$$\Delta f^* = MLP(\mathcal{F}_{\widetilde{\mathcal{L}\&\mathcal{G}}} \oplus Posenc(T) \oplus (flow + P)). \tag{7}$$

Adding $\Delta f^*$ to the current frame accurately predicts the point cloud at the target time step.

### 3.7 Objective Function

The proposed NeuroGauss4D-PCI optimizes the weights of 4D neural fields and Gaussian deformation fields using three self-supervised losses, eliminating the need for labeled data. We employ a cumulative temporal loss to refine network parameters, facilitating efficient spatiotemporal modeling of 4D point clouds while ensuring smoothness and distribution consistency.

**Chamfer Distance Loss**: Measures the bidirectional distance between the predicted point cloud $P_1^*$ and the ground truth $P_2$, promoting proximity between the predicted and ground truth points.

$$\ell_{CD}(P_1^*, P_2) = \sum_{p_1^* \in P_1^*} \min_{p_2 \in P_2} \|p_1^* - p_2\|_2^2 + \sum_{p_2 \in P_2} \min_{p_1^* \in P_1^*} \|p_2 - p_1^*\|_2^2. \tag{8}$$

**Smoothness Constraint Loss**: Enhances the smoothness of the predicted inter-frame flow field $\Delta f^*$, where $N(\Delta f_i^*)$ denotes the neighbors of $\Delta f_i^*$. It encourages similarity in flow vectors among neighboring points, ensuring smoother transformations in large-scale scenes. This loss is not required when modeling Dynamic Human Bodies.

$$\ell_{Smooth}(\Delta f^*) = \sum_{\delta f_i^* \in \Delta f^*} \frac{1}{|N(\delta f_i^*)|} \sum_{\delta f_j^* \in N(\delta f_i^*)} \left\|\Delta f^*(\delta f_j^*) - \Delta f^*(\delta f_i^*)\right\|_2^2. \tag{9}$$

**Earth Mover's Distance**: Calculates the minimum cost to move points from $P_1^*$ to match the distribution of $P_2$, where $\mathbb{T}_{P_1 \to P_2}$ is the optimal transport map. It promotes a distribution in the predicted point cloud that resembles the ground truth.

$$\ell_{EMD}(P_1^*, P_2) = \min_{\mathbb{T}_{P_1^* \to P_2}} \frac{1}{N} \sum_{p_1^* \in P_1^*} \|p_1^* - \mathbb{T}_{P_1^* \to P_2}(p_1^*)\|_2^2. \tag{10}$$

The overall loss integrates these functions across all time steps $t_i, t_j$ and temporal frames $b_i, b_j$, weighted by $\lambda_1, \lambda_2, \lambda_3$. It jointly optimizes the geometric precision, smoothness, and distribution similarity of the predicted point clouds across multiple time steps and temporal frames.

$$\mathcal{L} = \sum_{t_i, t_j \in T} \sum_{b_i, b_j \in B} \left(\lambda_1 \ell_{CD}(P_{b_i}^{t_i}, P_{b_j}^{t_j}) + \lambda_2 \ell_{Smooth}(\Delta f_i^{t_i \to t_j}) + \lambda_3 \ell_{EMD}(P_{b_i}^{t_i}, P_{b_j}^{t_j})\right). \tag{11}$$

Table 1: Quantitative comparison with open-source methods on DHB-Dataset [2]. Errors are scaled by $\times 10^{-3}$ to emphasize small-scale differences in human body metrics. "↓" means lower is better. "↑" means higher is better. Red and blue denote the first and second best metrics, respectively.

| Methods | Longdress | | Loot | | Red&Black | | Soldier | | Squat | | Swing | | Overall | | Param. ↓ |
|---|---|---|---|---|---|---|---|---|---|---|---|---|---|---|---|
| | CD | EMD | CD | EMD | CD | EMD | CD | EMD | CD | EMD | CD | EMD | CD ↓ | EMD ↓ | |
| IDEA-Net [2] | 0.89 | 6.01 | 0.86 | 8.62 | 0.94 | 10.34 | 1.63 | 30.07 | 0.62 | 6.68 | 1.24 | 6.93 | 1.02 | 12.03 | – |
| PointINet [8] | 0.98 | 10.87 | 0.85 | 12.10 | 0.87 | 10.68 | 0.97 | 12.39 | 0.90 | 13.99 | 1.45 | 14.81 | 0.96 | 12.25 | 1.30M |
| NSFP [11] | 1.04 | 7.45 | 0.81 | 7.13 | 0.97 | 8.14 | 0.68 | 5.25 | 1.14 | 7.97 | 3.09 | 11.39 | 1.22 | 7.81 | 0.12M |
| PV-RAFT [10] | 1.03 | 6.88 | 0.82 | 5.99 | 0.94 | 7.03 | 0.91 | 5.31 | 0.57 | 2.81 | 1.42 | 10.54 | 0.92 | 6.14 | **0.11 M** |
| NeuralPCI [3] | 0.70 | 4.36 | 0.61 | 4.76 | 0.67 | 4.79 | 0.59 | 4.63 | 0.03 | 0.02 | 0.53 | 2.22 | 0.54 | 3.68 | 1.85 M |
| Ours | 0.68 | 3.69 | 0.59 | 4.12 | 0.65 | 4.20 | 0.57 | 4.14 | 0.00 | 0.00 | 0.00 | 0.00 | 0.42 | 2.69 | 0.10 M |

Table 2: Quantitative comparison with other advanced methods on the NL Drive dataset. Frame-1, Frame-2, and Frame-3 denote three interpolated frames evenly spaced between two middle input frames. The symbol † signifies outlier removal during preprocessing (Sec. 3.2).

| Methods | Type | Frame-1 | | Frame-2 | | Frame-3 | | Average | |
|---|---|---|---|---|---|---|---|---|---|
| | | CD | EMD | CD | EMD | CD | EMD | CD ↓ | EMD ↓ |
| NSFP [11] | Forward Flow | 0.94 | 95.18 | 1.75 | 132.30 | 2.55 | 168.91 | 1.75 | 132.13 |
| | Backward Flow | 2.53 | 168.75 | 1.74 | 132.19 | 0.95 | 95.23 | 1.74 | 132.05 |
| PV-RAFT [10] | Forward Flow | 1.36 | 104.57 | 1.92 | 146.87 | 1.63 | 169.82 | 1.64 | 140.42 |
| | Backward Flow | 1.58 | 173.18 | 1.85 | 145.48 | 1.30 | 102.71 | 1.58 | 140.46 |
| PointINet [8] | Bi-directional Flow | 0.93 | 97.48 | 1.24 | 110.22 | 1.01 | 95.65 | 1.06 | 101.12 |
| NeuralPCI [3] | Neural Field | 0.72 | 89.03 | 0.94 | 113.45 | 0.74 | 88.61 | 0.80 | 97.03 |
| Ours | 4D Gaussian Deformation | 0.70 | 86.90 | 0.93 | 112.1 | 0.72 | 88.85 | 0.78 | 95.95 |
| Ours† | 4D Gaussian Deformation | 0.64 | 71.92 | 0.88 | 91.9 | 0.65 | 72.16 | 0.72 | 78.66 |

## 4 Experiments

### 4.1 Implementation Details

**Datasets:** Dynamic Human Bodies (DHB) dataset [2] contains 14 sequences of non-rigid human motions. The NL-Drive [3] dataset captures large-scale motion scenes from KITTI [32], Argoverse 2 [33], and Nuscenes [34] for autonomous driving. KITTIs [35; 32] and KITTIo [36; 32] are two versions of the KITTI Scene Flow dataset with and without occlusion masks, respectively.

**Metrics:** Like existing methods [3; 2; 8], we use Chamfer Distance (CD) and Earth Mover's Distance (EMD) to measure the consistency between predicted and ground truth point clouds. For the 3D scene flow task, we employ two error metrics $EPE_{3D}$ and $Outliers$ and two accuracy metrics, $ACC_S$ and $ACC_R$, to quantify performance.

**Experimental Setup:** We sample the input points to 1024 for object-level scenes and 8192 for autonomous driving scenes. NeuroGauss4D-PCI consists of 5 components, as shown in Table 4. Refer to the *supplementary materials* for the parameter settings of each component and more experimental details.

### 4.2 Result Comparison on Point Cloud Interpolation

Table 1 demonstrates the superior performance of the proposed method, consistently outperforming others on CD and EMD metrics, achieving near-zero errors for sequences like Squat and Swing. The dynamic soft Gaussian representation effectively models non-rigid deformations in human motions by transforming raw point clouds into multiple Gaussian distributions, with radial basis functions learning their temporal residuals. Fusing 4D neural field features preserves per-point detail. The extremely low errors on the DHB dataset validate the model's excellence in non-rigid motion modeling and point cloud interpolation. Notably, the proposed method achieves the best performance with significantly fewer parameters (0.09M) compared to methods like PointINet [8] (1.30M) and NeuralPCI [3] (1.85M). This is attributed to the efficient Gaussian representation and deformation field design, enabling superior performance through a more compact model.

On large-scale autonomous driving LiDAR datasets [3], NeuroGauss4D-PCI demonstrates exceptional temporal point cloud prediction/interpolation performance, significantly outperforming uni-directional

Table 3: Comparison of our method with the best-performing methods on multiple datasets and metrics. 'Self', and 'Full' represent self-supervised, and supervised methods, respectively. Unlike other methods inputs that use adjacent point clouds $(P_1, P_2)$, NeuroGauss4D-PC utilizes $(P_1, t_2)$.

| Method | Sup. | Inference Input | KITTIs [35; 32] | | | | KITTIo [36; 32] | | | |
|---|---|---|---|---|---|---|---|---|---|---|
| | | | $EPE_{3D}\downarrow$ | $ACC_S\uparrow$ | $ACC_R\uparrow$ | $Outliers\downarrow$ | $EPE_{3D}\downarrow$ | $ACC_S\uparrow$ | $ACC_R\uparrow$ | $Outliers\downarrow$ |
| FlowNet3D [36] | Full. | $(P_1, P_2)$ | 0.1767 | 37.38 | 66.77 | 52.71 | 0.183 | 9.8 | 39.4 | 79.9 |
| PointPWC [37] | Full. | $(P_1, P_2)$ | 0.0694 | 72.81 | 88.84 | 26.48 | 0.118 | 40.3 | 75.7 | 49.6 |
| Bi-PointFlow [38] | Full. | $(P_1, P_2)$ | 0.0300 | 92.00 | 96.00 | 14.10 | 0.065 | 76.9 | 90.6 | 26.4 |
| PT-FlowNet[39] | Full. | $(P_1, P_2)$ | 0.0224 | 95.51 | 98.38 | 11.86 | – | – | – | – |
| GMSF [40] | Full. | $(P_1, P_2)$ | 0.0215 | 96.22 | 98.50 | 9.84 | **0.033** | 91.6 | 95.9 | 13.7 |
| SCOOP$^+$ [42] | Self. | $(P_1, P_2)$ | 0.0390 | 93.60 | 96.50 | 15.20 | 0.047 | 91.3 | 95.0 | 18.6 |
| SPFlowNet [41] | Self. | $(P_1, P_2)$ | 0.0362 | 87.24 | 95.79 | 17.71 | 0.086 | 61.1 | 82.4 | 39.1 |
| Ours | Self. | $(P_1, t_2)$ | **0.0190** | **97.72** | **99.48** | **9.43** | 0.035 | **94.8** | **97.5** | **11.1** |

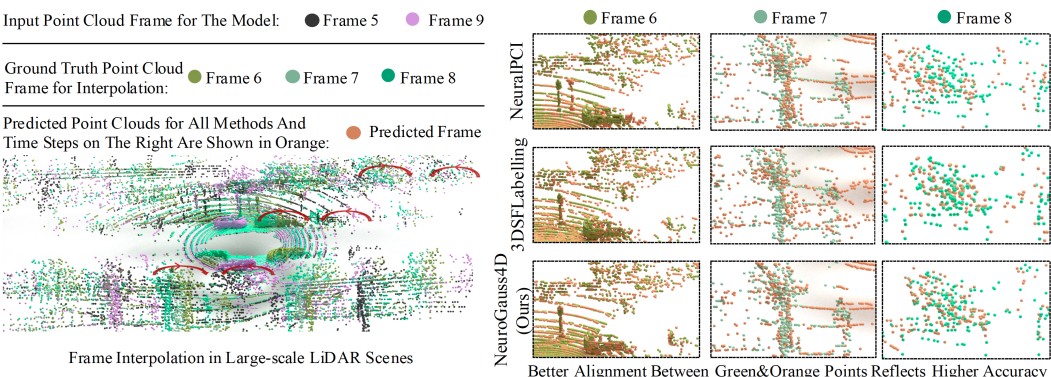

Figure 5: Compared to advanced point cloud interpolation algorithms [3; 9], our method aligns better with the ground truth in predicting point cloud positions and geometry on the NL Drive autonomous driving dataset [33; 34; 32; 3].

3D flow [11; 10] and neural field method [3], as shown in Table 2. It effectively handles challenges in temporal LiDAR scenes, such as large-scale non-linear motions, occlusions, and non-uniform data distributions (Fig. 1, and Fig. 5). NeuroGauss4D-PCI models the 4D temporal point cloud as a Gaussian deformation field with continuous and differentiable Gaussian representations, ensuring smooth interpolation and robust predictions. Temporal Gaussian graph convolutions capture local and global spatio-temporal correlations, enabling fine-grained motion prediction. Notably, Gaussian representations offer parameter efficiency, expressing complex geometric shapes and non-linear motions smoothly with fewer parameters compared to neural field methods.

## 4.3 Result Comparison on Point Cloud Scene Flow

To demonstrate the effectiveness of the proposed point cloud interpolation model in capturing complex deformations and transient motion patterns between two frames, we evaluated its performance on 3D scene flow estimation using the Scene Flow KITTI dataset (both non-occluded [35; 32] and occluded [36; 32]), using error and accuracy metrics. As shown in Table 3, our method outperformed approaches based on feature pyramids (PointPWC [37], Bi-PointFlow [38]), complex 3D point cloud transformers (PT-FlowNet [39], GMSF [40]), and self-supervised learning (SPFlowNet [41], SCOOP$^+$ [42]) across multiple metrics. Unlike other methods that require learning inter-frame correspondences, NeuroGauss4D-PCI only needs a single point cloud and target timestamp to predict the corresponding point cloud during inference. In the learning phase, NeuroGauss4D-PCI accurately models complex non-rigid deformations and details in temporal point clouds by integrating latent neural fields, iterative Gaussian representations, and 4D deformation fields. Its compact iterative Gaussian representations significantly reduce parameters while effectively capturing continuous point cloud changes by learning temporal residuals of Gaussian distribution parameters through radial basis functions, exhibiting superior capability in inter-frame point cloud matching.

Table 4: Ablation study of the proposed components on DHB [2] and NL-Drive [3]. The components include: Neural Field for learning spatio-temporal features and iterative Gaussian representation of 3D point plouds ( Gauss.PC )(Sec.3.2); T-RBF-GR represents Temporary Radial Basis Function Gaussian Residual Module (Sec.3.3); 4D Deformation for modeling 4D deformation fields (Sec.3.4); LG-Cat and Fast-LG-Fusion are different latent geometric feature fusion methods (Sec.3.5).

| Components | | | | | | DHB ($\times 10^{-3}$) | | NL-Drive | | Param. |
| Neural Field | Gauss.PC | T-RBF-GR | 4D Deformation | LG-Cat | Fast-LG-Fusion | CD↓ | EMD↓ | CD↓ | EMD↓ | |
|---|---|---|---|---|---|---|---|---|---|---|
| ✓ | | | | | | 0.58 | 3.70 | 1.06 | 105.43 | 0.027M |
| ✓ | ✓ | | | | | 0.57 | 3.62 | 1.03 | 103.58 | 0.028M |
| | ✓ | ✓ | | | | 0.59 | 3.81 | 1.04 | 107.86 | 0.029M |
| | ✓ | | ✓ | | | 0.50 | 3.04 | 0.80 | 98.57 | 0.092M |
| | ✓ | ✓ | ✓ | | | 0.49 | 2.99 | 0.80 | 98.03 | 0.093M |
| ✓ | ✓ | ✓ | ✓ | ✓ | | **0.42** | **2.69** | **0.79** | **97.36** | 0.099M |
| ✓ | ✓ | ✓ | ✓ | | ✓ | **0.44** | **2.48** | **0.78** | **95.95** | 0.103M |

## 4.4 Ablation Study

Table 4 shows the effect of each component on model performance. Using just the neural field, the Chamfer Distance (CD) measures $0.58 \times 10^{-3}$ on the DHB dataset and 1.06 on the NL-Drive dataset, showcasing its capacity to capture spatio-temporal features. Adding the Gaussian point cloud representation slightly lowers the CD by 0.01 and 0.03 on DHB and NL-Drive, respectively, indicating limited benefits from this integration. Removing the neural field and relying solely on the Gaussian point cloud and T-RBF-GR module significantly worsens performance, emphasizing the neural field's critical role in modeling 4D spatio-temporal point clouds. However, integrating the 4D Gaussian Deformation Field with the Gaussian representation markedly enhances performance, with a 14% and 19% decrease in CD and a 17% and 6.5% reduction in Earth Mover's Distance (EMD) on the two datasets, respectively. This highlights the deformation field's effectiveness in accurately representing dynamic variations. Adding the T-RBF-GR module further improves performance, showcasing its utility in addressing dynamics and temporal correlations. By fusing the latent features and the explicit 4D deformation field features, we effectively reduce errors in point cloud prediction. The Fast-LG-Fusion module, in particular, achieves the best performance, attaining the lowest CD and EMD across datasets, except for a slight 0.02 increase in CD on the DHB dataset compared to direct concatenation.

## 5 Conclusion

NeuroGauss4D-PCI is proposed for accurate 4D temporal point cloud modeling and interpolation. 4D neural fields encode latent spatio-temporal features, and an iterative Gaussian cloud representation structures point clouds. A temporal RBF Gaussian residual module smoothly updates Gaussian distribution parameters. 4D Gaussian deformation fields leverage temporal graph convolutions for robust dynamic point cloud modeling. Efficient fusion of latent spatio-temporal and robust Gaussian deformation features enables 3D point cloud prediction. Evaluated on multiple datasets, the method demonstrates significant superiority over existing approaches, particularly in complex dynamics scenarios. Ablation studies validate the contribution of each module to overall performance.

**Broad Implications:** Our work is the first to successfully combine radial basis functions for predicting 3D Gaussian distribution residuals and introduce 4D Gaussian deformation fields into the temporal point cloud prediction task. As the supplementary materials indicate, this is a widely applied yet underexplored issue. Finally, we hope that the proposed NeuroGauss4D-PCI can provide valuable assistance to the community in exploring temporal point cloud understanding and modeling.

**Limitations:** 1) Interpretability: The integration of various features and the opacity of deep neural networks pose challenges in understanding the decision-making process and the fundamental principles behind model predictions. 2) Efficiency: Similar to NeRF [43], nearly 90% of the time is consumed by runtime scene optimization, with inference accounting for only about 10% of the time.

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

# Supplemental Material

## A  Datasets

Dynamic Human Bodies (DHB) [2] dataset consists of point cloud sequences of 14 non-rigid deformable human motions. Six sequences—Longdress, Loot, Redandblack, Soldier, Squat 2, Swing are designated for testing, and the remaining eight for training. The NL-Drive dataset [3], integrating data from KITTI odometry [32], Argoverse 2 sensors [33], and Nuscenes [34], focuses on capturing large-scale motion scenes and is divided into training, validation, and test sets in a 14:3:3 ratio. It specifically targets hard samples with significant inter-frame motion and large self-motion, employing detailed 6-DOF pose transformations and a filtering mechanism to support the training and testing of advanced autonomous driving algorithms. KITTIs [35; 32] and KITTIo [36; 32], two versions of the KITTI Scene Flow dataset processed for 3D point cloud scene flow evaluation, exclude occluded points in the former, containing 142 training scenes, while the latter includes all points with occlusion masks, featuring 150 training scenes.

## B  Method Details

Table 5: Model Input and Hyperparameter Settings

| Component | Setting |
|---|---|
| Model Input | 4 frames of point clouds sampled at regular 3-frame intervals: 
 - DHB dataset: 1024 points per frame 
 - NL-Drive dataset: 8192 points per frame 
 Time step 0 to 1, spatial and temporal dimensions: 3 and 1 |
| Iterative Gaussian Cloud Soft Clustering | - Number of Gaussian components $M$: 16 (LiDAR), 8 (DHB) 
 - Number of iterations $\kappa$ for fitting Gaussian ellipsoids: 200 
 - DGCNN for local geometric feature learning: 
 5 convolutional blocks with kernel size 1, batch normalization, 
 ReLU activation, and dropout (0.3 probability) 
 Input channels: $16 \times 2$, $16 \times 2$, $16 \times 2$, $16 \times 2$, $16 \times 4$ 
 Output channels: 16, 16, 16, 16, 64 
 Dynamic k-NN with $k$ ranging from 8 to 32 |
| 4D Neural Field | - Input MLP: width 1280, depth 1 
 - Hidden MLPs: 2 with width 32, depth 5 
 - Decoder MLP: width 32, depth 1 
 - Leaky ReLU activation |
| Temporal RBF Gaussian Residual | - 4 RBF centers (for 4 time steps) 
 - Input feature dimension: 32 
 - RBF centers evenly spaced between 0 and 1 
 - Learnable standard deviations for RBFs 
 - Learnable translation, rotation, scale parameters for Gaussian ellipsoids 
 - Learnable transformation parameters for input features |
| 4D Gaussian Deformation Field | - Temporal encoding dimension: 8 
 - Intermediate and output dimensions for TGCN: 32 |
| Optimization 

 Weighted loss | - AdamW optimizer, learning rate 0.001, weight decay 0 
 - 5000 iterations, early stopping on optimal solution 
 - chamfer distance (1.0), earth mover's distance (50.0), 
 smoothness regularization (1.0) 
 - Poly learning rate scheduler |

**Input Setting:** For dynamic point cloud sequence evaluation, the input comprises 4 point cloud frames sampled at regular 3-frame intervals. Each frame contains 1024 points for the DHB dataset and 8192 points for the NL-Drive dataset. The temporal resolution is 1, with the initial frame at

time step 0 and the final frame at time step 1. The spatial and temporal dimensions are 3 and 1, respectively.

**Iterative Gaussian Cloud Soft Clustering:** A key hyperparameter is the number of Gaussian components $M$, set to 16 for LiDAR and 8 for DHB. The iteration count $\kappa$ for Gaussian ellipsoid parameter fitting is 200. A 5-block DGCNN learns local geometric features from the input Gaussian cloud data. Each block employs a 2D convolution (kernel size 1), batch normalization, ReLU activation, and dropout (0.3 probability). The input channels for blocks 1-4 are $16 * 2$, with 16 output channels each. The final block has $16 * 4$ input channels and 32 output channels. For graph representation, DGCNN uses k-NN with $k$ ranging from 8 to 32 based on input size.

**4D Neural Field:** The core comprises an input MLP (width 1280, depth 1), two hidden MLPs (width 32, depth 5), and a decoder MLP (width 32, depth 1), utilizing Leaky ReLU activation.

**Temporal Radial Basis Function Gaussian Residual:** Hyperparameters include 4 RBF centers (corresponding to time steps) and 32-dimensional input features. RBF centers are evenly distributed between 0 and 1, with learnable standard deviations. Translation, rotation, scale parameters of Gaussian ellipsoid distributions, and input feature transformation parameters are learnable.

**4D Gaussian Deformation Field:** Temporal information is encoded as an 8-dimensional vector. Intermediate and output dimensions for temporal Gaussian graph convolutional network are 32.

**Optimization:** AdamW optimizer (learning rate 0.001, weight decay 0), 5000 iterations, minimizing a weighted loss (chamfer distance loss weight 1.0, earth mover's distance loss weight 50.0, smoothness regularization loss weight 1.0), with early stopping. Poly learning rate scheduler is employed.

**Computational Performance:** The average training time is $\sim 1.31$ seconds per iteration, and the average evaluation time is $\sim 0.23$ seconds per frame. The GPU memory consumption is 7436 MiB. The model is evaluated on an NVIDIA GeForce RTX 3090 GPU with 24GB of VRAM. The implementation is based on the PyTorch deep learning framework, utilizing CUDA acceleration for efficient parallel computation on the GPU.

## C   Applications

Heterogeneous sensors like LiDAR and cameras often suffer from asynchronous data acquisition [44; 45], posing challenges in fusing their complementary data streams. As illustrated in Fig. 6, NeuroGauss4D-PCI offers a powerful solution by enabling the interpolation of arbitrary point cloud frames from continuous point clouds, facilitating precise multi-sensor time synchronization. This capability unlocks seamless integration of data from multiple sensors, enhancing the robustness and accuracy of perception systems in dynamic environments.

Moreover, NeuroGauss4D-PCI can be effectively applied to 4D automatic annotation tasks, assigning point-wise labels to unlabeled intermediate frames based on sparse 3D labels. This automated labeling process significantly reduces the time and effort required for manual annotation, a critical bottleneck in many computer vision applications. Consequently, it accelerates the development and deployment of systems relying on labeled point cloud data.

Furthermore, NeuroGauss4D-PCI demonstrates remarkable accuracy in LiDAR point cloud densification, generating high-resolution, dense point clouds from sparse inputs. Accurate dense point clouds serve as invaluable data sources for a wide range of autonomous driving perception tasks [46; 47; 48; 49], including depth estimation [50; 51], occupancy prediction [52], and 4D scene reconstruction [53; 54; 55]. The ability to obtain high-fidelity representations of the environment from limited sensor data highlights the immense potential of NeuroGauss4D-PCI in enabling robust and reliable perception systems.

## D   Supplementary Visualization

In the qualitative comparison (Fig. 7) on the DHB dataset [2], NeuroGauss4D-PCI demonstrates superior performance compared to state-of-the-art open-source models, including PointINet [8], IDEA-Net [2], and NeuralPCI [3], in the task of temporal point cloud interpolation. The proposed method excels in capturing and modeling non-rigid human motions with high fidelity, as evident from the visual results. NeuroGauss4D-PCI represents the input point cloud as a set of Gaussian ellipsoids

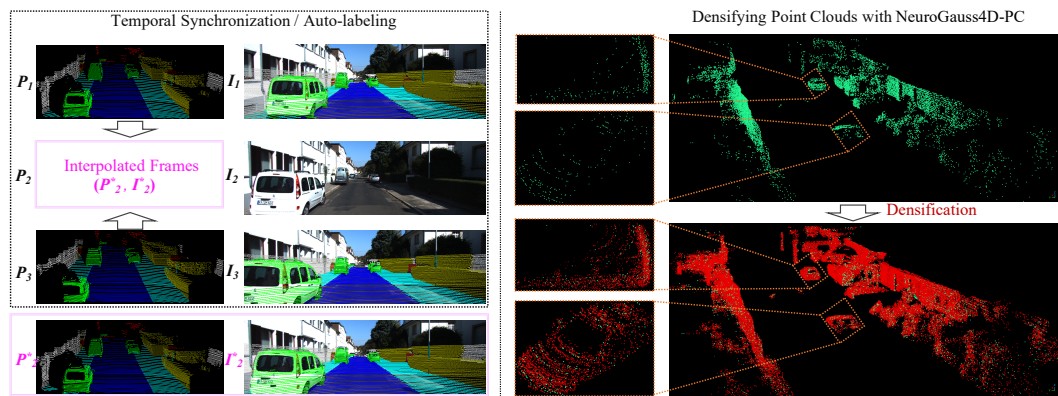

Figure 6: Visualization of NeuroGauss4D-PCI for multi-sensor time synchronization and point cloud densification applications. In point cloud densification, sparse ground truth points are shown in green, while the predicted dense point cloud is shown in red, exhibiting good overlap with the sparse ground truth.

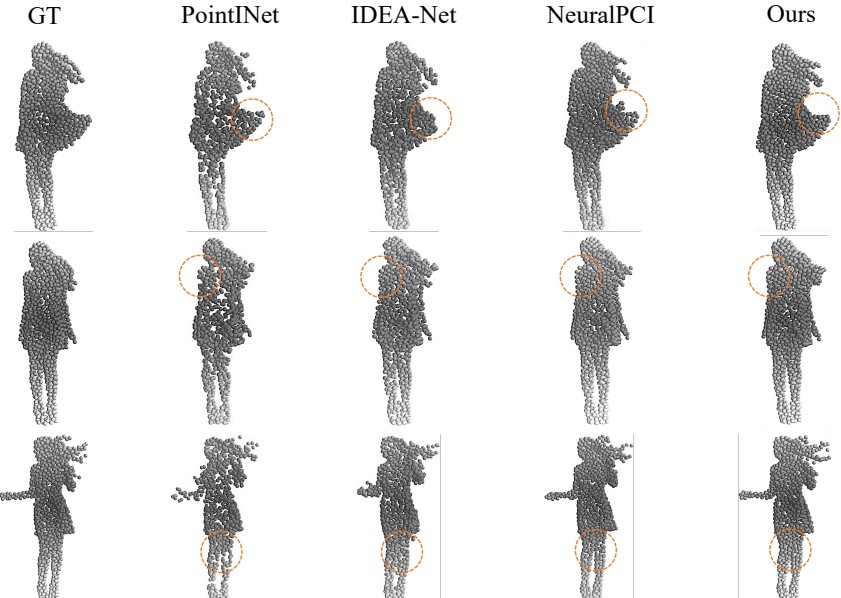

Figure 7: Qualitative visualizations on the DHB dataset demonstrate the evident superiority of the proposed method in reconstructing fine details when compared to existing state-of-the-art open-source models [37; 2; 3].

through the Iterative Gaussian Cloud Soft Clustering module, effectively encoding local geometric structures and establishing meaningful correspondences across temporal frames. Furthermore, the Temporal Radial Basis Function Gaussian Residual module enables effective learning of temporal dynamics and residual deformations by incorporating radial basis functions (RBFs) centered at different time steps, allowing for accurate interpolations, especially in scenarios involving complex non-rigid motions. The 4D Neural Field module, the core of NeuroGauss4D-PCI, leverages the expressive power of neural fields to model the continuous 4D space of point cloud sequences, encoding both spatial and temporal information in a unified neural field to generate high-quality interpolated frames that maintain spatial and temporal coherence. Additionally, the Feature Interpolator module employs an attention-based mechanism to aggregate and interpolate features from neighboring time steps, enabling cross-temporal propagation and integration of relevant information, further enhancing the capability to handle non-rigid deformations. Overall, NeuroGauss4D-PCI's innovative

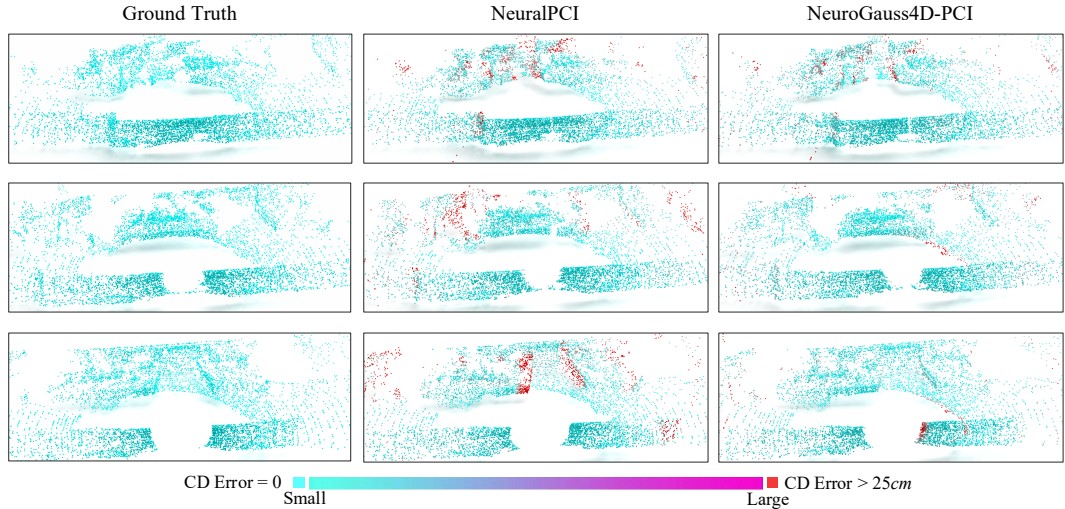

Figure 8: Quantitative comparison with pure neural field method [3] on the NL-Drvie dataset.

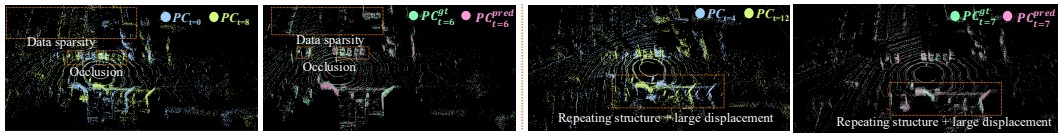

Figure 9: Point cloud interpolation in challenging autonomous driving scenarios. Blue/yellow: input clouds at $T_{0,4}/T_{8,12}$. Green: ground truth at $T_{6,7}$. Pink: predictions at $T_{6,7}$. The scenes feature local occlusions, sparse point clouds, repeating structures, and large displacements simulated by extended frame intervals. The overlap between green and pink points demonstrates our algorithm's accuracy in these complex scenarios, showcasing its robustness to data sparsity, occlusions, ambiguous temporal features from repeating structures, and large environmental displacements.

architecture, combining Gaussian ellipsoid representations, temporal RBF residuals, 4D neural fields, and attention-based feature interpolation, endows it with a unique ability to accurately model and interpolate non-rigid point cloud sequences. The qualitative results on the DHB dataset demonstrate the method's superiority in handling complex human motions.

The qualitative visualization (Fig. 8) on the NL-Drive dataset showcases NeuroGauss4D-PCI's exceptional performance in point cloud interpolation and prediction for autonomous driving scenarios. Compared to NeuralPCI, our method exhibits superior temporal consistency and local coherence, benefiting from the robust mathematical constraints imposed by the Gaussian parameterization. For unstructured and sparse point clouds, the 4D modeling approach enables more accurate predictions in sparse and unstructured regions. Additionally, NeuroGauss4D-PCI demonstrates robustness in handling occluded areas, as evident in the second row of the visualization.

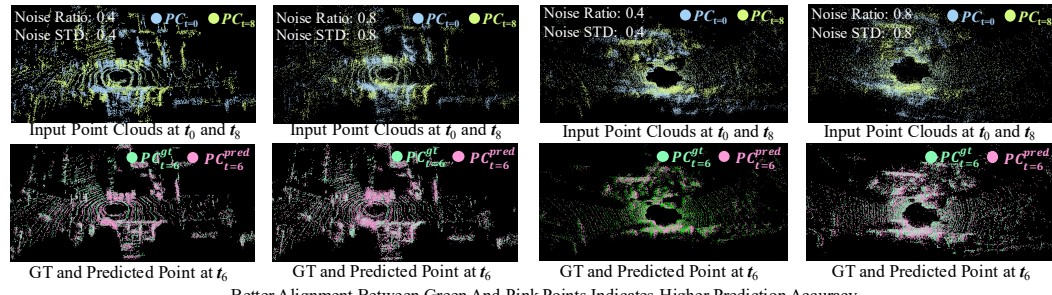

Figure 10: Robustness analysis of our point cloud interpolation method under varying noise conditions. Top row: Input point clouds $PC_{t=0}$ (blue) and $PC_{t=8}$ (yellow) with different noise ratios (0.4, 0.8) and standard deviations (0.4, 0.8). Bottom row: Comparison between ground truth $PC_{t=6}^{gt}$ (green) and predicted $PC_{t=6}^{pred}$ (pink) point clouds at $t = 6$. Noise is added to input point clouds using a Gaussian distribution, where noise ratio determines the proportion of points affected, and noise STD defines the standard deviation of the noise. The alignment between green and pink points indicates prediction accuracy. Results demonstrate our method's resilience to increasing noise levels, maintaining reasonable performance even under severe noise conditions (noise ratio 0.8, STD 0.8).

