# OpenReview forum: "NeuroGauss4D-PCI: 4D Neural Fields and Gaussian Deformation Fields for Point Cloud Interpolation"
_NeurIPS.cc/2024/Conference — NeurIPS 2024 poster_

### Official Review · Reviewer_2Dkp · 2024-07-09

**Soundness:** 3
**Presentation:** 1
**Contribution:** 3
**Rating:** 6
**Confidence:** 4

**Summary:**

Point Cloud Interpolation (PCI) is the task of predicting intermediate point cloud features from a sparser representation, often to construct point cloud representations for intermediate time frames. Contrary to interpolation in other fields, point clouds are largely unstructured and do not preserve consistent information across different sequence frames, and thus cannot be addressed adequately with classical interpolation functions such as linear interpolation. To properly represent point clouds in a latent space for learned approaches, a method that respects the volumetric nature of point distributions is crucial for minimising noisy artifacts caused by the unordered nature of point clouds.

This paper proposes three new components of PCI learning architectures that address this necessity. First, a Gaussian clustering function is used to group point sets into distinct regions, which are less prone to misrepresenting unintended variations in point cloud data, and facilitates a necessary structure for PCI. To my understanding, with the sparse latent representations, interpolation is achieved by first using Radial Basis Functions (RBF) to provide an initial smooth estimate of the interpolated Gaussian cluster sequence, followed by a learned graph-based network to determine a final interpolation based on the composition of RBF sequences from every cluster.

**Strengths:**

The paper is technically sound with strong results against state-of-the-art methods in PCI of dynamic environmental LiDAR scenes. The methodology presentation is largely clear and well structured. A detailed ablation study is provided to demonstrate the importance of each component for effective PCI based on standard Chamfer Distance and Earth Mover Distance metrics.

**Weaknesses:**

Section 1: Introduction
- The introduction writing structure is quite unconventional, and repeats itself word for word in the contributions statement. I suggest the introduction of the proposed method be explained in more detail with regards to its overall architectural structure, to incorporate a consistent flow with the presentation of the method's components.
- The mentioned challenges of PCI are sporadic and inconsistent with the way PCI is introduced as a research problem. The use of a mathematical formula to introduce PCI takes away from the intended motivation of PCI-centric research, and is therefore difficult to link to each challenge. e.g. If solving Eq.(1) is the goal, what does the representation of "multiple unordered point clouds" have to do with it? This is unclear with the current presentation.
- Figure 1 is a mess of graphs and qualitative examples that belong as separate (sub)figures in the Experiments section. The introduction of Figure 1 in the beginning of the work is inappropriate. What is the expected ground truth for the right side figures? It is unclear what the right side figure is trying to demonstrate.

Section 2: Related works
- Like the introduction, the presented explanation is centred around explaining mathematical formulae, rather than intuitively presenting technical problems and expressing it as formulae if necessary. It is generally unnecessary for the formula to be understood in order to analyse the current state of research for each subsection. For this reason, by linking the motives of past research to solving an otherwise arbitrary formula, the flow of the related works section is unclear.
- L81: "for efficient:" appears to be in error.

Section 3: Methodology
- 1), 2), 3) are not labelled in the diagram itself, and thus can be difficult to associate.
- Figure 3 fails to visualise the purpose of each component, i.e. RBF Activator, Gaussian/Feature Interpolators, in the proposed method. It is generally expected that a flowchart is used to both illustrate the information flow of the system, and provide labels for the purpose, i.e. I/O, of each component. Mathematical equations are highly ineffective for this.

**Questions:**

1. Figure 2: Is it correct to call the output of the soft-clustering process "spheres"? The figure appears to describe "ellipsoids" rather than consistent radius spheres.
2. Outlier removal in Section 4.2/Table 2 appears to improve the results of the method quite significantly. How does it affect other state-of-the-art methods?

**Limitations:**

Yes, limitations of the method are discussed briefly in the manuscript regarding its applicability in practical scenarios.

---

> ### Author Rebuttal · Authors · 2024-08-06
>
> We appreciate your meaningful suggestions and questions, and we will address each of them in our response.
> ## Q1.1: Non-traditional writing structure of the introduction.
> A1.1: We will rigorously revise our introduction structure according to your recommendations. Due to space constraints, please refer to the end of our response to A1 for the detailed revision plan.
> ## Q1.2: Sporadic PCI challenges and unclear logical relationship to Formula 1.
> A2:
> We appreciate your feedback and will clarify that "multiple unordered point clouds" refers to the inherently unordered nature of input point cloud signals. To enhance clarity and logical flow, we will separate the introduction of PCI concepts from the discussion of current challenges in the main text. Additionally, we will provide a detailed explanation for each term in Equation 1.
> ## Q1.3: Inappropriate introduction of Figure 1 at the beginning.
> A3:
> We greatly appreciate your valuable feedback on Figure 1. We agree that introducing detailed experimental results at the beginning of the paper may not be ideal.
>
> Our original intent was to quickly showcase our method's advantages, including:
> 1. Interpolation accuracy across various frame intervals
> 2. Performance with different data sparsity levels
> 3. Effectiveness on both human body and autonomous driving datasets
>
> To address your concern, we will:
> 1. Split the existing Figure 1
> 2. Move qualitative and quantitative results to the experimental section
> 3. Ensure readers fully understand our method before encountering these results
>
> Regarding the right side of Figure 1, we will provide further explanations in our PDF response (Fig. 4).
>
> The following is the revised Introduction outline, and we will re-revise the text strictly according to your suggestions:
> 1. Point Cloud Interpolation (PCI)
>    - Concept and application scenarios, incorporating Equation 1
>    - Clear explanation of each term in Equation 1
>
> 2. PCI Challenges and Research Significance
>    - Point cloud data characteristics and resulting challenges
>    - Complexity of spatio-temporal dynamic modeling
>    - Difficulty in generalizing from sparse temporal samples
>
> 3. NeuroGauss4D-PCI: Key Components and Their Roles
>
>    a. Iterative Gaussian Cloud Soft Clustering
>       - Structured representation for unordered point clouds
>       - Geometric feature capture addressing data irregularity
>
>    b. Temporal RBF-GR Module
>       - Complex non-linear temporal dynamics modeling
>       - Smooth interpolation between sparse temporal samples
>
>    c. 4D Gaussian Deformation Field
>       - Long-term motion trend and non-rigid deformation capture
>       - Cross-frame geometric consistency maintenance
>
>    d. 4D Neural Field
>       - Fine-grained spatiotemporal detail representation
>       - Multi-scale dynamic modeling complement to Gaussian representation
>
>    e. Fast Latent-Geometric Feature Fusion Module
>       - Adaptive combination of Gaussian and neural features
>       - Handling of varying point densities and enhanced spatiotemporal modeling
>
> 4. Innovations and Contributions
>    - Novel 4D representation combining Gaussian soft clustering and neural field features
>    - Continuous time modeling via Temporal RBF-GR module
>    - Advanced 4D Gaussian deformation field using graph convolutions
>
> ## Q2: Unclear flow in the related works section
> A2: We will improve the related works section by:
> 1. Reorganizing the structure of each subsection (neural fields, 3D Gaussian splatting, and point cloud interpolation)
> 2. Reducing reliance on mathematical formulas
> 3. Correcting grammatical errors
> ## Q3.1: Unlabeled components in Figure 2
> A3.1: We will revise Figure 2's caption to match the module names in the diagram and avoid using numbering:
>
> Figure 2: NeuroGauss4D-PCI architecture. The system processes input point clouds (X, Y, Z, T) and a predicted time T_pred. Latent Feature Learning occurs via Fourier Basis Encoding and Neural Field. Gaussian Representation is achieved using Iterable Gaussian SoftClustering. Temporal Modeling employs the RBF-GR Module. The 4D Deformation Field integrates latent features and temporal information. Feature Fusion is performed by the Fast-LG-Fusion module. Finally, Point Cloud Generation is accomplished using a Prediction Head. This pipeline enables effective spatiotemporal modeling and interpolation of point clouds.
> ## Q3.2: Figure 3 fails to visualize component purposes
> A3.2: We've enhanced Figure 1 in the PDF reply as follows:
>
> 1. Gaussian Ellipsoid Visualization:
>    Added pre- and post-interpolation Gaussian ellipsoids, showing model inputs and outputs.
>
> 2. Intermediate Process Visualization:
>    Included visualizations of intermediate products to illustrate data transformation.
>
> 3. Component Functionality:
>    Clarified inputs, outputs, and functions of RBF-GR module components.
>
> 4. Information Flow Diagram:
>    Will add a flow diagram in the final revision, annotating each component's role and data flow.
> ## Q4: Incorrect terminology in Figure 2.
> A4: Using "sphere" to describe soft clustering output is indeed inaccurate.. We will:
>
> - Change "Gaussian spheres" to "Gaussian ellipsoids" in Figure 2's caption and explanation
> - Review and correct all related descriptions in the main text
>
> ## Q5: Impact of outlier removal on other methods
>
> A5: Outlier removal improved results for all tested methods. Here's a comparison:
>
> | Methods   | CD ↓   | EMD ↓   |
> |-----------|--------|---------|
> | NSFP      | 1.75   | 132.13  |
> | NSFP†     | 1.72   | 129.05  |
> | NeuralPCI | 0.80   | 97.03   |
> | NeuralPCI† | 0.75  | 83.56   |
> | Ours      | 0.78   | 95.95   |
> | Ours†     | **0.72**   | **78.66**   |
>
> Note: † indicates results with outlier removal.
>
> NeuroGauss4D-PCI benefits most from outlier removal due to:
> 1. Our Gaussian representation amplifies the benefits of outlier removal.
> 2. Clean data significantly enhances our 4D deformation field's accuracy.
> 3. Outlier removal maximizes our Temporal RBF-GR module's effectiveness.

---

> > ### Comment · Reviewer_2Dkp · 2024-08-08
> >
> > Thank you for the response. The new figures, explanations, and results are much needed improvements to the original submission. I am inclined to increase my rating in order to reflect the technically sound and adept contributions of the paper, provided that its presentation is acceptable in the final version.

---

> > > ### Author Response · Authors · 2024-08-08
> > > **Response to Reviewer 2Dkp**
> > >
> > > Thank you very much for your response and valuable feedback. We deeply appreciate your recognition of our revised paper. Your constructive comments have greatly improved the quality of our research.
> > >
> > > We will do our utmost to further refine the presentation of our paper according to your suggestions, ensuring that the final version meets your expectations.
> > >
> > > Once again, thank you for your time and insights.

---

### Official Review · Reviewer_ddrR · 2024-07-12

**Soundness:** 4
**Presentation:** 3
**Contribution:** 3
**Rating:** 8
**Confidence:** 3

**Summary:**

The paper presents a novel model, “NeuroGauss4D-PCI,” for point cloud frame interpolation, a popular but challenging 3D computer vision task in real-world scenarios such as Lidar point cloud densification. The model outperforms existing PCI methods on the most popular benchmark datasets, DHB and NL-Drive, showcasing scalability to tasks like auto-labeling and point cloud densification.

**Strengths:**

-	The paper is well written and well-organized, a good review of related works supports the motivation, the experiment is well designed, and the results and additional information are satisfied.
-	Although I’m not an expert in developing such algorithms in detail, as a user from the application side, I can understand that using 4D Gaussian deformation fields and temporal radial basis function Gaussian residual modules to capture complex spatiotemporal dynamics is a novel approach.

**Weaknesses:**

-	There are no details of the experiment platform or discussion of the computational efficiency of the proposed method.
-	The potential of NeuroGauss4D-PCI seems significant. But as we all know, NeuralPCI was said to be the best model for PCI until now. A more detailed comparison with this model could be crucial in providing valuable insights and strengthening the paper's contribution to the field.

**Questions:**

I don’t have additional detailed questions for this paper, as I may not be good enough to understand all the details of the proposed algorithm.

**Limitations:**

I wonder if it is convincing that all are testing their work on only two open datasets used in this paper.

---

> ### Author Rebuttal · Authors · 2024-08-06
>
> We sincerely appreciate your positive feedback on our paper's organization, literature review, experimental design, and results. Your recognition of our novel approach using 4D Gaussian deformation fields and temporal radial basis function Gaussian residual modules to capture complex spatiotemporal dynamics is particularly encouraging. We are grateful for your detailed comments and will address each of your concerns in the following responses.
>
> ## Q1: Discussion on computational efficiency
> A1:
>  - Experimental Platform:
> Our algorithm was tested on a platform equipped with an NVIDIA RTX 3090 GPU.
>
>  - Computational Efficiency:
> We recorded detailed computational costs for different point cloud sizes:
>
> Table 1: Time consumption statistics (in seconds)
> | Processing Step                            | 1024 points | 8192 points |
> |:-------------------------------------------|------------:|------------:|
> | **Single Frame**                           |             |             |
> | Time Encoding                              |      0.0003 |      0.0003 |
> | 4D Neural Field                            |      0.0007 |      0.0008 |
> | RBF-GR+4DGD                                |      0.0041 |      0.0034 |
> | LG-fusion                                  |      0.0004 |      0.0028 |
> | Prediction Head                            |      0.0005 |      0.0023 |
> | Loss Calculation                           |      0.0048 |      0.0022 |
> | **Total (Single Frame)**                   |  **0.0108** |  **0.0118** |
> | **Sequence (4 Frames)**                    |             |             |
> | Loss Backpropagation + Optimizer Update    |      0.0567 |      0.0590 |
> | **Total (One Sequence Iteration)**         |  **0.0753** |  **0.1572** |
>
>  - Efficiency Analysis
> 1. Single frame processing: Only 0.0118 seconds for 8192 points, demonstrating high efficiency.
> 2. Sequence processing: 0.1572 seconds for a 4-frame sequence (8192 points/frame), or about 157.2 seconds for 1000 iterations.
> 3. Preprocessing optimization: Iterative Gaussian Cloud Soft Clustering module moved to preprocessing stage, running once per sequence (~1 second) instead of every iteration.
> 4. Weight freezing: DGCNN and SelfAttention network weights frozen to reduce updatable parameters.
>
> ## Q2: Detailed comparison with NeuralPCI.
> A2:
> We appreciate the reviewer's valuable feedback and recognition of NeuroGauss4D-PCI's potential. As requested, we have expanded our comparison with NeuralPCI:
>
> 1. Visual comparison in Figures 1, 6 and 7 in the original paper.
> 2. Quantitative comparisons across various point cloud densities, frame intervals, datasets, and metrics are presented in Tables 1 and 2, and Figure 1 in the original paper.
>
> Key advantages of NeuroGauss4D-PCI over NeuralPCI include:
>
> 1. Structured Representation: Our iterative Gaussian soft clustering module provides a structured temporal point cloud representation, better capturing geometric features and spatio-temporal dynamics. NeuralPCI, in contrast, directly inputs spatial and temporal coordinates to an MLP.
>
> 2. Spatio-temporal Modeling: Our Temporal RBF-GR module and 4D Gaussian deformation fields enable more precise modeling of complex non-rigid deformations and non-linear trajectories, particularly beneficial for long time spans and complex dynamic scenes.
>
> 3. Feature Fusion: Our fast latent-geometry fusion module adaptively combines implicit features from neural fields with explicit geometric features from Gaussian deformation fields, enhancing spatio-temporal correlation modeling.
>
> These improvements collectively contribute to NeuroGauss4D-PCI's superior performance in various challenging scenarios.
>
> ## Q3: Convincingness of testing on only two open datasets.
> A3:
> Thank you for raising this important question. I understand your concerns about the potential limitations of testing only on two open datasets, as well as the reliability of the experimental design and results of all the work.
>
> 1. Dataset Representativeness:
>    - DHB (object-level) and NL-Drive (large-scale autonomous driving) represent two major application scenarios in point cloud interpolation.
>    - NL-Drive combines KITTI, Argoverse, and Nuscenes datasets, offering diverse, real-world autonomous driving environments.
>    - These datasets cover a wide range from fine-grained object deformations to complex, large-scale scenes.
>
> 2. Fair Comparison:
>    - We benchmark against state-of-the-art methods including IDEA-Net, PointINet, NSFP, PV-RAFT, NeuralPCI, and 3DSFLabelling.
>    - We use results provided by Zheng et al. in the NeuralPCI paper, validating their experimental outcomes for consistency.
>
> 3. Open-source Verification:
>    - All compared algorithms are open-sourced, enhancing result credibility and reproducibility.
>    - We conducted independent experiments using NSFP and 3DSFLabelling's open-source code on the test set, further ensuring fair and accurate comparisons.
>
> 4. Extensibility Validation:
>    - We demonstrate NeuroGauss4D-PCI's potential in related tasks such as auto-labeling and point cloud upsampling, showcasing its versatility and adaptability.
>
> This comprehensive approach aims to establish the reliability and broad applicability of our method across diverse scenarios in point cloud interpolation.

---

> > ### Comment · Reviewer_ddrR · 2024-08-08
> >
> > Thanks for the detailed response and the additional experiment. I am satisfied with the current version. and my recommendation will not change from the previous one.

---

> > > ### Author Response · Authors · 2024-08-08
> > > **Response to Reviewer ddrR**
> > >
> > > Thank you very much for your valuable feedback and recognition of our work. We will do our best to improve the quality of our work based on your suggestions, ensuring that the final version meets your expectations.
> > > Finally, thank you again for your time and insights.

---

### Official Review · Reviewer_Z54i · 2024-07-14

**Soundness:** 3
**Presentation:** 1
**Contribution:** 2
**Rating:** 4
**Confidence:** 4

**Summary:**

The paper introduces NeuroGauss4D-PCI, a model designed to address the challenges of point cloud interpolation (PCI) by using Gaussian soft clustering and a 4D neural field to model complex non-rigid deformations in dynamic scenes. The model excels in capturing spatial and temporal dynamics from sparse data, showing superior performance on object-level and large-scale datasets.

**Strengths:**

It is interesting to utilize Gaussian clustering and 4D neural fields for solving PCI problems.

**Weaknesses:**

1. The color scheme of the figures in the article is very poor, making many details difficult to discern.

2. Point cloud interpolation is a relatively niche direction in point cloud analysis. The author should more clearly emphasize the significance of studying this issue.

3. The most related work reported in this work might be the CVPR 23 paper Neuralpci.  Does that mean no more recent related works?

**Questions:**

1. How does NeuroGauss4D-PCI ensure robustness against varied data sparsity in real-world scenarios?

2. Can the model handle rapidly changing dynamic scenes, such as explosions or quick animal movements?

3. What are the computational costs associated with training and inference, and how do they compare to simpler models?

4. How does the model perform under conditions of significant occlusion or sensor noise?

**Limitations:**

1. The complex interactions between Gaussian fields and neural networks may hinder understanding and tuning of the model.

2. The high computational demand may limit deployment in time-sensitive applications.

---

> ### Author Rebuttal · Authors · 2024-08-06
>
> We sincerely thank the reviewer for their insightful questions and provide detailed responses below
> ## Q1: Poor color scheme.
> A1:
> We commit to redesigning the color scheme in our main text to improve visibility and readability. For your reference, we've implemented this new color scheme in our PDF response.
> ## Q2: Emphasize the significance of research direction.
> A2: We will reemphasize the research significance of this task in our paper:
> 1. Bridges the gap between current sensor limitations and high-frequency data demands in autonomous driving and AR.
> 2. Improves object tracking and motion prediction in dynamic environments.
> 3. Facilitates world model construction and AI-generated 3D content.
> 4. Supports multi-sensor synchronization, auto-labeling, and point cloud densification.
>
> Our supplementary materials demonstrate these applications, highlighting PCI's versatility and potential across various 3D vision tasks.
> ## Q3: Latest relevant works.
> A3:
> NeuralPCI (CVPR 2023) is the most recent specialized study on point cloud interpolation in our citations. While 3DSFLabelling (CVPR 2024) addresses related tasks, it focuses on 3D scene flow rather than sequence interpolation. Recent applications of point cloud interpolation include skeletal animation generation (PC-MRL, arXiv 2024) and optimal transport evaluation (CL+SW2, WACV 2024).
> ## Q4: Ensuring robustness to different data sparsities.
> A4:
> NeuroGauss4D-PCI achieves robust performance across data sparsity levels due to:
>
> 1. Adaptive Representation: Gaussian soft clustering extracts meaningful features even from sparse data.
>
> 2. Flexible Temporal Modeling: RBF-GR module interpolates smoothly between frames, regardless of point density.
>
> 3. Multi-scale Capture: 4D Gaussian fields model both global structure and local details, adapting to varying sparsity.
>
> These components synergistically handle different data densities. Our algorithm's robustness across diverse point cloud densities is demonstrated by these results:
>
> | Point Cloud Size | NSFP | PV-RAFT | PointINet | NeuralPCI | 3DSFLabelling | Ours† |
> |:----------------:|:----:|:-------:|:---------:|:---------:|:------------:|:----:|
> |       1024       | 6.81 |  6.92   |   5.64    |   5.04    |     6.19     | **4.01** |
> |       2048       | 4.01 |  3.98   |   3.10    |   2.59    |     3.41     | **2.00** |
> |       4096       | 2.60 |  2.51   |   1.70    |   1.38    |     1.80     | **0.99** |
> |       8192       | 1.80 |  1.65   |   1.10    |   0.80    |     1.19     | **0.72** |
> |      16384       | 1.30 |  1.21   |   0.72    |   0.48    |     0.92     | **0.38** |
>
> ## Q5: Handling rapidly changing dynamic scenes？
> A5:
> In our response PDF (Fig. 2), we present visualizations of frame interpolation predictions for scenarios involving long intervals and rapid motion patterns. Our predicted results demonstrate strong alignment with ground truth.
>
> ## Q6: Training and inference costs comparison.
> A6:
> Our model's computation costs for different point cloud sizes are detailed below:
>
> | Processing Step                            | 1024 points | 8192 points |
> |:-------------------------------------------|------------:|------------:|
> | **Single Frame**                           |             |             |
> | Time Encoding                              |      0.0003 |      0.0003 |
> | 4D Neural Field                            |      0.0007 |      0.0008 |
> | RBF-GR+4DGD                                |      0.0041 |      0.0034 |
> | Fast LG-fusion                             |      0.0004 |      0.0028 |
> | Prediction Head                            |      0.0005 |      0.0023 |
> | Loss Calculation                           |      0.0048 |      0.0022 |
> | **Total (Single Frame)**                   |  **0.0108** |  **0.0118** |
> | **Sequence (4 Frames)**                    |             |             |
> | Loss Backpropagation + Optimizer Update    |      0.0567 |      0.0590 |
> | **Total (One Sequence Iteration)**         |  **0.0753** |  **0.1572** |
>
> Note: The Iterative Gaussian Cloud Soft Clustering module runs only once in the preprocessing stage, taking about 0.2248 seconds, and is not included in the table.
>
> For 8192 points, our model takes 0.1572 seconds per iteration, or about 157.2 seconds for 1000 iterations. Simpler models like NeuralPCI take about 60 seconds for 1000 iterations, while more complex models like 3DLSFLabelling take over 10 minutes for a single point cloud pair. Our algorithm's time consumption is within an acceptable range for optimization-based methods designed for accuracy.
>
> ## Q7: Performance under occlusion or sensor noise?
> A7:
> Figures 2 and 3 in our response PDF demonstrate our model's visual results under significant occlusion and various noise conditions, respectively. The results show that our model maintains acceptable accuracy even in these challenging scenarios.
> ## Q8: Complex interactions between modules.
> A8:
> Our experiments demonstrate that integrating Gaussian fields with neural network features consistently yields performance improvements, regardless of the specific fusion method employed:
>
> | Method        | DHB (×10⁻³)           | NL-Drive               |
> |:--------------|:----------------------|:-----------------------|
> |               | CD ↓    | EMD ↓      | CD ↓    | EMD ↓        |
> | Baseline      | 0.49    | 2.99       | 0.80    | 98.03        |
> | LG-Cat        | 0.42    | 2.69       | 0.79    | 97.36        |
> | Fast-LG-Fusion| 0.44    | 2.48       | 0.78    | 95.95        |
>
> The combination provides:
> Gaussian fields capture spatial structure and global information, while neural networks learn complex non-linear mappings and local features.
>
> ## Q9: High computational cost limiting real-time applications.
> A9:
> As shown in the runtime analysis in Answer 6, our algorithm's computational cost falls within an acceptable range. Our approach, similar to NeRF, is a coordinate-based, per-scene fitting method primarily designed for offline optimization of individual scenes.

---

> > ### Author Response · Authors · 2024-08-13
> >
> > Dear Reviewer Z54i,
> >
> > We greatly appreciate your time and expertise in reviewing our work. Your insightful feedback is invaluable for improving our paper. In our rebuttal, we have:
> >
> > - Addressed all your concerns
> > - Improved figure clarity
> > - Emphasized research significance
> > - Highlighted recent related work
> >
> > Our response demonstrates NeuroGauss4D-PCI's effectiveness across various scenarios through:
> >
> > - Visualizations
> > - Tables
> > - Detailed computational cost analysis
> >
> > **With only about 20 hours left in the author-reviewer discussion period, we kindly request your feedback. Your thoughts on any remaining concerns are crucial for our paper's improvement.**
> >
> > Thank you sincerely for your time and effort.
> >
> > Best regards,
> >
> > The Authors of Paper #7315

---

### Official Review · Reviewer_ZfCH · 2024-07-16

**Soundness:** 3
**Presentation:** 3
**Contribution:** 2
**Rating:** 5
**Confidence:** 4

**Summary:**

This paper proposes a novel method for point cloud interpolation, which aims to address complex non-rigid scenarios. It turns point clouds into 3D Gaussians via iterative soft clustering, and then utilizes several 4D spatio-temporal modules to fuse latent features with neural fields. Specifically, this paper employs the temporal radial basis function for Gaussian interpolation, the graph convolutional network for feature extraction and the attention module for feature fusion. Comprehensive experiments demonstrate the leading performance across indoor and outdoor datasets, along with a thorough ablation study for the proposed components.

**Strengths:**

- It is the first to introduce 3D Gaussians for point cloud interpolation, which shows great performance in modeling non-rigid deformations and non-linear trajectories.

- The proposed 4D Gaussian deformation fields leverage temporal graph convolutions for spatio-temporal feature aggregation.

- The experiments are conducted on multiple datasets, including indoors and outdoors. It is also compared on the scene flow benchmark to demonstrate its effectiveness. And the ablation study is comprehensive for each proposed module.

**Weaknesses:**

- The proposed method is overly complex and redundant, which shows a trivial combination of existing modules and limited contribution. Moreover, the proposed modules lack strong motivation and necessity, and the interpretability of the intermediate features is difficult to clarify.
- According to the Supplementary Material, nearly two hours of optimization time is required for just four frames of input, which is much more than the previous method. The problem may lie in the design of proposed method, which is not suitable for an optimization-based approach, but rather a learning-based approach. For example, some feature extraction or fusion may not be meaningful or necessary for a single sample fitting.

**Questions:**

- This paper may not be related to the work of 3DGS series. It just uses 3D Gaussians but has nothing to do with "splatting". Also, the proposed method is more similar to the super-point concept or the Gaussian Mixture Model in clustering and should be discussed.
- Does DGCNN use pre-trained models or is it trained from scratch? Retraining DGCNN for each optimization can be time consuming and may not be meaningful for fitting individual samples.
- In the Temporal RBF-GR Module, the 3D Gaussian needs to be interpolated, so how to ensure the correspondence? Also, is it feasible to visualize the Residual Gaussian to prove its reasonableness and validity?
- Quantitative comparisons of efficiency with other methods. It would be better to analyze the training/inference time for each module. In addition, can the proposed method be extended to more frames or more points of input, and what is the corresponding efficiency?



typo:

- L136 - "Table4.2"?

- missing arrows of ACC_S and ACC_R in Table 3

**Limitations:**

- The entire framework can be further simplified and improve its efficiency.
- Additional supervision or constraints can be introduced for 3D Gaussians rather than only points.
- Use more challenging datasets beyond DHB. In Table 1, the results for both Squat and Swing sequences are 0.00, which may need to retain higher precision or change to other datasets.

---

> ### Author Rebuttal · Authors · 2024-08-06
>
> Thank you for your insightful critique.
> ## Q1: Algorithm complexity and limited contributions
> A1:
> **Due to space constraints, we kindly refer you to Reviewer xq5Q's Q1 for the explanation on model complexity. We apologize for any inconvenience.**
>
> **Design Motivation:**
> - Gaussian Soft Clustering: Structures multiple unordered point clouds, improving upon methods like NeuralPCI
> - Temporal RBF-GR Module: Overcomes linear motion assumptions of methods such as PointINet
> - 4D Gaussian Deformation Fields: Captures complex motion patterns over extended time spans, surpassing scene flow-based approaches
>
> **Key Innovations**
>  - 4D Gaussian representation for spatiotemporal features
>  - Temporal RBF-GR for smooth parameter interpolation
>  - 4D Gaussian Graph Convolutional Deformable module for spatiotemporal learning
>
> **Improved Interpretability**
>    - Uses explicit geometric principles: Gaussian means (μ), covariance matrices (Σ), deformation fields
> ## Q2: Long optimization time and suitability for optimization-based approach
> A2:
> Following your Q4 suggestion, we've significantly reduced the algorithm's runtime.
>
> Our core components suit optimization methods because:
>
> **Temporal RBF-GR, 4D Gaussian Deformation**: Optimize meaningful mathematical parameters.
>
> **4D Neural Field**: Expresses highly complex non-linear functions through optimization.
>
> ## Q3: Relationship to 3D Gaussians, splatting, and clustering
> A3:
> Relationship to 3DGS:
>  - Uses 3D Gaussians, inspired by 3DGS concepts
>  - Doesn't employ "splatting" techniques
>  - Will reduce 3DGS discussion in the paper
>
> Our method uses EM-like clustering and GMM concepts, enhanced by DGCNN and self-attention for Gaussian deformation, focusing on point cloud interpolation. Detailed relationships will be clarified in the main text.
>
> ## Q4: DGCNN training approach
> A4:
>  - Moved Iterative Gaussian Cloud Soft Clustering to preprocessing stage
>  - Froze weights of DGCNN and SelfAttention networks
>  - Networks now act as fixed feature extractors
>
> **Performance Impact**
>    - Model with frozen weights performs comparably to original model
>
> | Methods               | Longdress |      | Loot   |      | Red&Black |      | Soldier |      |
> |-----------------------|-----------|------|--------|------|-----------|------|---------|------|
> |                       | CD        | EMD  | CD     | EMD  | CD        | EMD  | CD      | EMD  |
> | NeuralPCI             | 0.70      | 4.36 | 0.61   | 4.76 | 0.67      | 4.79 | 0.59    | 4.63 |
> | Ours                  | **0.68**  | **3.69** | **0.59** | 4.12 | **0.65** | **4.20** | **0.57** | 4.14 |
> | Ours (Freeze weight)  | 0.69      | 3.92 | **0.59** | **4.03** | 0.66    | 4.51 | **0.57** | **4.13** |
>
> **Efficiency Gains**
>    - Processing time for 4-frame sequence (8192 points) reduced to 0.1572 seconds.
>    - Iterative Gaussian Cloud Soft Clustering now takes about 0.2248 seconds in preprocessing
> For detailed timing of individual model components, please refer to our response to Reviewer xq5Q's Q9.
>
> ## Q5: 3D Gaussian correspondence during interpolation, and residual Gaussian visualization.
> A5:
> Reason:
> 1. Each Gaussian is uniquely identified and tracked across all timesteps.
> 2. Gaussian parameters evolve smoothly over time via RBF interpolation.
> 3. Self-attention mechanism ensures coherent global deformation.
>
> Residual Gaussian visualization provided in Figure 1 of response PDF.
> ## Q6: Model efficiency analysis for more frames/points, and writing corrections
> A6:
> 1. Detailed timing for each module provided in reviewer xq5Q's A9 for 1024 and 8192 points.
> 2. For optimization time comparisons across models with varying frame counts and point cloud sizes:
>
> | Method        | 4 Frames |         |         | 6 Frames |         |         | 8 Frames |         |         |
> |:--------------|:--------:|:-------:|:-------:|:--------:|:-------:|:-------:|:--------:|:-------:|:-------:|
> |               | 1024 pts | 8192 pts| 16384 pts| 1024 pts | 8192 pts| 16384 pts| 1024 pts | 8192 pts| 16384 pts|
> | 3DSFLabelling | 0.1324   | 0.3620  | 0.7500  | 0.1986   | 0.5430  | 1.1250  | 0.2648   | 0.7240  | 1.5000  |
> | NeuralPC      | 0.0447   | 0.0861  | 0.1924  | 0.0671   | 0.1292  | 0.2886  | 0.0894   | 0.1722  | 0.3848  |
> | Ours          | 0.0753   | 0.1572  | 0.3948  | 0.1130   | 0.2358  | 0.5922  | 0.1506   | 0.3144  | 0.7896  |
>
> The time consumption of our algorithm is within a reasonable range.
>
> 3. Writing corrections: "Table 4.2" reference on line 136 and missing arrows for ACC_S and ACC_R in Table 3.
> ## Q7: Framework simplification for improved efficiency
> A7: Framework simplified as described in A4, significantly improving efficiency.
> ## Q8: Additional supervision for 3D Gaussian distributions
> A9:
> Explored two additional constraints:
>
> **Gaussian Distribution Consistency Constraint**: Similar to the Smoothness Loss in the paper, we enforced continuity and consistency between Gaussian distributions of adjacent time steps.
>
> **Local Anisotropy Constraint**: We quantified the similarity of local directional and structural features by comparing covariance matrices of corresponding local regions in target and predicted point clouds.
>
> Results:
> | Constraint | Dataset | CD | EMD |
> |------------|---------|----:|----:|
> | Baseline | DHB | 0.44 | 2.48 |
> | | NL-Drive | 0.78 | 95.95 |
> | Gaussian Consistency | DHB | 0.43 | 2.53 |
> | | NL-Drive | 0.79 | 95.06 |
> | Local Anisotropy | DHB | 0.44 | 2.39 |
> | | NL-Drive | 0.79 | 93.22 |
>
> Precision remains nearly constant, but at the cost of increased computation time.
>
> ## Q9: Evaluation on more challenging datasets
>
> A10:
> 1. Evaluated on NL Drive dataset (KITTI, Argoverse, Nuscenes) in Table 2 of paper.
> 2. Additional testing on Waymo dataset:
>
> | Method         |   CD |   EMD |
> |----------------|-----:|------:|
> | 3DSFLabelling  | 0.72 | 83.07 |
> | NeuralPC       | 0.51 | 60.94 |
> | Ours           | **0.48** | **50.71** |
>
> These results further validate our model's effectiveness in complex, real-world autonomous driving scenarios.

---

> > ### Comment · Reviewer_ZfCH · 2024-08-12
> >
> > Thanks for your efforts during the rebuttal, which addressed most of my concerns. For the efficiency improvements that make a big difference from the original version, I hope that the authors will clearly modify the relevant content in the final version and include as much as possible the additional issues mentioned in the review and the corresponding experiments. Based on this, I would keep my positive rating.

---

### Official Review · Reviewer_xq5Q · 2024-07-22

**Soundness:** 3
**Presentation:** 2
**Contribution:** 2
**Rating:** 5
**Confidence:** 4

**Summary:**

This paper proposes a method, NeuroGauss4D, to tackle the problem of 4d point cloud interpolation. NeuroGauss4D consists of the following 5 components:
1. "Iterative Gaussian Soft Clustering": a module to encode the scene to Gaussian representation with DGCNN features (map: X, Y, Z, T -> Gaussians(Mu, Cov, Feat)).
2. "4D Neural Field": a neural field with Fourier features and MLP to capture spatio-temporal features in latent space (map: X, Y, Z, T -> latent).
3. "Temporal RGB-GR": a module with temporal radial basis functions to interpolate Gaussian parameters in continuous time (map: Mu, Cov, Feat, T -> ΔMu, ΔCov, ΔFeat, ΔT).
4. "4D Gaussian Deformation Field": a module that take Gaussian parameters (Mu, Cov, Feat) and the latent features as input to predict the motion and feature deformation field (map: Mu, Cov, Feat, T -> deformation field).
5. "Fast-LG-Fusion": an attention mechanism that fuses the latent features (FL) and the geometric features (FG, from the deformation field) to predict the point cloud at the target time step (map: FL, FG, T -> point cloud).

**Strengths:**

1. This method shows the application of dynamic Gaussian representation to the problem of 4D point cloud interpolation.
2. This method outperforms previous methods on the set of evaluations provide by the author.

**Weaknesses:**

1. This method shows the application of dynamic Gaussian representation to the problem of 4D point cloud interpolation, which is a good idea. However, the execution of the idea seems to make the method overly complex. NeuroGauss4D-PCI's pipeline contains 5 different components, where a mixture of representations are used. Besides more hand-tuning for hyperparameters and longer training time, this complexity could make it hard to interpret the fundamental principles and the key components behind the full model.
2. It seems that the method is not very efficient. For example, the "Iterable Gaussian Soft Clustering" procedure shown in Algorithm 1 is only capable of taking one timestamp of point cloud (N, 3) as input, which means that this has to be done for each timestamp of the point cloud sequence.
3. Similar to the point above, the method might be limited on the size of the point cloud sequence that it can handle. For example, line 218 says "we sample the input points to 1024 for object-level scenes and 8192 for autonomous driving scenes". A point cloud of 1024 points in autonomous driving scenes is very small, which could limit the method's applicability to real-world scenarios. The authors can provide more information on how the method will perform when we have a larger point cloud size or a longer sequence of point clouds.
4. The author provides rich information about the method, which is good, but the clarity of the writing can be improved. Some of these are mentioned in the "Questions" section, and it would be nice to include them in future versions of the paper. There are other inconsistencies in the paper, for example:
    - Equation 1 uses 0-based indexing, but line 29 uses 1-based indexing.
    - Line 109 has an extra double quote.
    - Line 117 says "Iterative Gaussian Cloud Soft Clustering" but the Figure 2 uses "**Iterable** Gaussian ...".
    - Equations shall be written as part of a sentence, not as separate sentences (Eq. 5, 8, 9, 10, 11; and also Eq. 4).

**Questions:**

1. It would be good to clarify in paper the exact meaning of "Frame-1, Frame-2, and Frame-3" in Table 2. Does it mean that we fit the scene with frame {0, 4, 8, 12, ...}, and then we evaluate on frame {1, 2, 3, 5, 6, 7, 9, 10, 11, ...}, and we call frame {1, 5, 9, ...} as Frame-1, frame {2, 6, 10, ...} as Frame-2, and frame {3, 7, 11, ...} as Frame-3?
2. In Table 5, when we say "4 frames of point clouds sampled at regular 3-frame intervals", does it mean that for each scene, we only train on 4 frames {0, 4, 8, 12} and we evaluate on 9 frames {1, 2, 3, 5, 6, 7, 9, 10, 11}? If this is the case, why do we only train on these 4 frames but not fit the full point cloud sequence of the scene in the raw dataset?
3. In line 456, when we say "initial frame at time step 0 and the final frame at time step 1", what do the "0" and "1" refer to? For example, in equation 1, it is clear that there can be time steps larger than 1, e.g. time step 4, 8, etc. Is the "0" and "1" in line 456 referring to some form of normalized time step? So, what exactly is the time step gap (in seconds) between the training frame and evaluation frames?
4. From equation 1, it seems that the method only fits one model for one whole scene, which is good. Can you confirm this?
5. In line 476 we report the "average training time is ∼ 1.31 seconds per iteration". Per my understanding, this "iteration time" means the training iteration. It is not clear if the time includes "Iterative Gaussian Cloud Soft Clustering" (which has to be done for each time step), and other feature extraction steps. It would be more clear to give a high-level timing number, for example, could you report "how long does it take to fit a full of ~X training frames each containing ~Y number of points"?

**Limitations:**

Yes, the authors have discussed the above.

---

> ### Author Rebuttal · Authors · 2024-08-06
>
> We appreciate your thorough review and valuable feedback. Here are our responses to your questions:
> ## Weaknesses
> ### Q1: Is the method overly complex with its 5 components?
> **A1:** Our method is designed for efficiency and effectiveness:
> 1. **Parameter Efficiency:** NeuroGauss4D-PCI uses only about 0.1 million parameters, significantly fewer than similar methods requiring millions.
> 2. **Logical Pipeline:** Our components form a clear sequence:
>
>    a) Structure point clouds
>
>    b) Model temporal evolution
>
>    c) Capture fine details
>
>    d) Fuse information
> 3. **Component Breakdown:**
>    - Gaussian Soft Clustering: Organizes unstructured point clouds
>    - Temporal RBF-GR Module: Enables smooth temporal interpolation
>    - 4D Gaussian Deformation Field: Simulates long-term motion trends
>    - 4D Neural Field: Captures high-frequency details
>    - Fusion Module: Combines geometric and latent features
> 4. **Benefits:**
>    - Improved Interpretability
>    - Reduced Parameter Count
> Each component addresses specific challenges in point cloud interpolation.
> ### Q2: Is it inefficient that the algorithm must perform operations on each timestamp of the point cloud sequence?
> **A2:** NeuroGauss4D-PCI is designed for per-scene optimization, not real-time processing.
> - Processes single-frame point clouds sequentially
> - Accumulates loss across the entire sequence
> - Performs backpropagation and optimization after processing all frames
>
> Single-frame processing takes about 10ms. The majority of computation time is spent on backpropagation and parameter updates after processing the full sequence. Nevertheless, optimizing a large-scale LiDAR scene requires only 2-3 minutes.
> ### Q3: Can the method handle large point cloud sizes or sequences?
> **A3:** We use different point cloud sizes for distinct scenarios:
> - Object-level scenes: 1024 points. (Aligns with Idea-net)
> - Autonomous driving scenes: 8192 points. (Consistent with NeuralPCI)
>
> We also provide quantitative comparisons of different point cloud densities:
>
> **Table 1: Quantitative comparison results on NLDrive dataset with different input point cloud densities.**
> | Point Cloud Size | NSFP | PV-RAFT | PointINet | NeuralPCI | 3DSFLabelling | Ours† |
> |:----------------:|:----:|:-------:|:---------:|:---------:|:------------:|:----:|
> |       1024       | 6.81 |  6.92   |   5.64    |   5.04    |     6.19     | **4.01** |
> |       2048       | 4.01 |  3.98   |   3.10    |   2.59    |     3.41     | **2.00** |
> |       4096       | 2.60 |  2.51   |   1.70    |   1.38    |     1.80     | **0.99** |
> |       8192       | 1.80 |  1.65   |   1.10    |   0.80    |     1.19     | **0.72** |
> |      16384       | 1.30 |  1.21   |   0.72    |   0.48    |     0.92     | **0.38** |
>
> ### Q4: How will you improve writing clarity?
> **A4:** We will:
> - Unify to 0-based indexing
> - Remove punctuation errors
> - Use consistent terminology
> - Integrate equations into sentences
>
> ## Questions
> ### Q5: Clarify "Frame-1, Frame-2, and Frame-3" in Table 2.
> **A5:** We use frames {0, 4, 8, 12} as input and evaluate on frames {5, 6, 7}.
> ### Q6: Explain the training and evaluation process in Table 5.
> **A6:** We optimize on frames {0, 4, 8, 12} and evaluate on {5, 6, 7}. We also present experimental results for both shorter and longer frame intervals:
>
> **Table 2: Comparison of CD errors (×10^-2^) for different methods across various time intervals**
> | Time Interval (frames) | NSFP | PV-RAFT | PointINet | IDEA-Net | NeuralPCI | 3DSFLabelling | Ours |
> |:----------------------:|:----:|:-------:|:---------:|:--------:|:---------:|:-------------:|:----:|
> | 1 | 0.69 | 0.78 | 0.91 | 1.01 | 0.49 | 0.69 | **0.39** |
> | 2 | 0.98 | 0.86 | 0.913 | 1.02 | 0.52 | 0.73 | **0.40** |
> | 3 | 1.22 | 0.90 | 0.92 | 1.03 | 0.53 | 0.89 | **0.42** |
> | 4 | 1.48 | 1.03 | 0.99 | 1.09 | 0.63 | 1.03 | **0.43** |
> | 5 | 1.87 | 1.30 | 1.132 | 1.192 | 0.72 | 1.12 | **0.46** |
> | 6 | 2.01 | 1.38 | 1.23 | 1.32 | 0.90 | 1.30 | **0.48** |
> | 7 | 2.20 | 1.502 | 1.41 | 1.48 | 1.06 | 1.48 | **0.51** |
> | 8 | 2.42 | 1.83 | 1.77 | 1.50 | 1.31 | 1.80 | **0.54** |
> | 9 | 2.83 | 2.02 | 1.81 | 1.76 | 1.48 | 1.99 | **0.57** |
>
> Our method shows the lowest CD error across all time intervals.
> ### Q7: Explain the time steps "0" and "1" in line 456.
> **A7:** We normalize all input times to 0-1 to handle different sequence lengths, simplify training, and enhance generalization.
> ### Q8: Does the method fit one model per scene?
> **A8:** Yes, NeuroGauss4D-PCI uses per-scene fitting, allowing high-quality interpolation at any time point after a 3-minute optimization.
> ### Q9: Can you provide detailed timing information?
> **A9:** Here's our time consumption breakdown:
> **Table 3: Algorithm Time Consumption Statistics**
> | Processing Step                            | 1024 points | 8192 points |
> |:-------------------------------------------|------------:|------------:|
> | **Single Frame**                           |             |             |
> | Time Encoding                              |      0.0003 |      0.0003 |
> | 4D Neural Field                            |      0.0007 |      0.0008 |
> | RBF-GR+4DGD                                |      0.0041 |      0.0034 |
> | LG-fusion                                  |      0.0004 |      0.0028 |
> | Prediction Head                            |      0.0005 |      0.0023 |
> | Loss Calculation                           |      0.0048 |      0.0022 |
> | **Total (Single Frame)**                   |  **0.0108** |  **0.0118** |
> | **Sequence (4 Frames)**                    |             |             |
> | Loss Backpropagation + Optimizer Update    |      0.0567 |      0.0590 |
> | **Total (One Sequence Iteration)**         |  **0.0753** |  **0.1572** |
>
> Note: The Iterative Gaussian Cloud Soft Clustering module runs only once in the preprocessing stage, taking about 0.2248 seconds, and is not included in the table. The total time for one sequence iteration includes the processing time for 4 single frames plus additional sequence-level operations.

---

> ### Comment · Reviewer_xq5Q · 2024-08-12
>
> I thank the author for the rebuttal responses, where most of my concerns are covered. Score increase 4->5.

---

> > ### Author Response · Authors · 2024-08-13
> >
> > Dear Reviewer xq5Q,
> >
> > Thank you for your consideration and the increased score. We appreciate your feedback and will incorporate your suggestions in our final paper.
> >
> > Best regards,
> >
> > Authors of Paper #7315

---

### Author Rebuttal · Authors · 2024-08-06

We sincerely thank the Area Chair for their time and effort in handling our paper, and all reviewers for their detailed and valuable suggestions, which are crucial for improving our work.

The reviewers have positively acknowledged our method's novelty, impact, accuracy, and potential, as highlighted below:

Our proposed methodology presents a novel approach to addressing the Point Cloud Interpolation (PCI) challenge, introducing 3D Gaussian representation for PCI (Reviewer **ZfCH**) and innovatively combining Gaussian clustering with 4D neural fields (Reviewer **Z54i**). The use of 4D Gaussian deformation fields and temporal radial basis function Gaussian residual modules is noted as an innovative method for capturing complex spatiotemporal dynamics (Reviewer **ddrR**). The method demonstrates excellent performance in non-rigid deformation and non-linear trajectory modeling (Reviewer **ZfCH**). The experimental design is considered comprehensive and reasonable, including evaluation on multiple indoor and outdoor datasets (Reviewers **ZfCH**, **2Dkp**) and detailed ablation studies (Reviewers ZfCH, 2Dkp). Results show our method outperforming existing approaches on standard evaluation metrics (Reviewers **xq5Q**, **2Dkp**). Additionally, the paper's organization, writing quality (Reviewers **ddrR**, **2Dkp**), and review of related work supporting the research motivation (Reviewer **ddrR**) are commended.

After careful analysis of all comments, the main issues can be summarized as:

1. Model structure complexity
2. Algorithmic efficiency of components
3. Model's adaptability to varying frame numbers and point cloud densities
4. Latest PCI research and its significance
5. Performance in complex scenarios (noise, rapid motion, occlusion)
6. Writing errors, structural clarity, and figure readability
7. Performance on challenging datasets and impact of outliers

We have meticulously revised the manuscript based on these valuable suggestions. Due to space constraints, we've provided concise responses here, with detailed illustrations and experimental results in the reviewer-specific replies.

Brief responses to the main issues:

1. While our model comprises multiple components, each addresses specific PCI challenges. We've clarified component relationships and demonstrated their effectiveness through ablation studies.
2. We provide detailed timing statistics for each component across various point cloud sizes.
3. Our model shows robust performance across different point cloud densities (1024 to 16384 points) and frame intervals, with comparative results demonstrating superior performance in various scenarios.
4. We've re-explained the latest PCI research and its specific significance.
5. Visualizations in our response demonstrate effective interpolation under occlusion, noise, and rapid motion conditions, maintaining good alignment with ground truth.
6. We will address all mentioned and potential writing and presentation issues, improve figure clarity, and resolve terminology inconsistencies to enhance overall readability and comprehension.
7. We've tested on challenging datasets like Waymo and provided comparative results. Outlier removal benefits all methods, with our approach showing the most significant improvements due to its Gaussian representation and temporal modeling capabilities.

We look forward to engaging in further discussions with you over the next few days. We are eager to address any remaining concerns you may have about our paper and are committed to providing comprehensive clarifications to ensure the highest quality of our research.

Once again, we extend our sincere gratitude to the Area Chair and all reviewers for their invaluable input and dedication!

---

### Decision · Program_Chairs · 2024-09-25

**Decision:**

Accept (poster)

**Comment:**

This paper proposes a novel method for point cloud interpolation using Gaussian soft clustering and a 4D neural field. The experimental results on multiple datasets, including both indoor and outdoor environments, demonstrate the effectiveness of the proposed method.

Four reviewers (2x BA, 1x WA, 1x SA) acknowledged the strengths of this paper, and the authors' rebuttal successfully addressed most of the concerns raised by these reviewers.

Reviewer Z54i, however, provided a negative rating of Borderline Reject, concerning issues with "the poor color scheme of the figures," "the significance of the research direction," and "robustness to more challenging scenes." The authors responded to these concerns, but no further feedback was received from the reviewer.

After reviewing the paper, the reviews, and the rebuttal discussion, the AC has decided to recommend acceptance. For the camera-ready version, please improve the color scheme in the figures and revise the writing to enhance visibility and readability. Additionally, include the new experiments and discussions to further strengthen the manuscript.